# Trajectory Graph Copilot: Pre-Action Error Diagnosis in LLM Agents

## Abstract

Large language model(LLM)-based agents have demonstrated exceptional performance across a wide range of complex interactive tasks. However, they often struggle with long-horizon interactive tasks common in domains like embodied AI. The complexity and vast action spaces in these settings lead to compounding errors, where a single suboptimal action can derail an entire trajectory, causing the agent to exhaust its limited step budget on inefficient or unrecoverable paths. To overcome this without costly fine-tuning, we draw inspiration from software debugging, where execution logs are analyzed to preemptively catch errors. We propose Trajectory Graph Copilot, a novel framework that acts as a "copilot" for LLM agents by diagnosing potential action errors before they are executed. At its core, Gebugger models historical trajectories as a probabilistic graph and uses a Graph Neural Network to identify sequential action patterns that frequently lead to failure. Functioning as a proactive diagnostic sandbox, our method provides early warnings on potentially flawed actions, prompting the agent to self-correct. This pre-action error diagnosis prevents costly mistakes, significantly enhancing the agent's ability to complete long-horizon tasks successfully. The extensive experiments on four benchmarks with three LLM agents demonstrate a $14.69\%$ pass ratio improvement on average.

## 1 Introduction

Large language models (LLMs), such as ChatGPT (OpenAI, 2022), Gemini (Team et al., 2024), and Llama (Touvron et al., 2023), possess a remarkable capacity language comprehension and generation. When equipped with tools (Yang et al., 2023; Wu et al., 2024a), these LLMs become powerful agents capable of extraordinary performance in complex applications such as coding (Islam et al., 2024; Qian et al., 2023; Zhang et al., 2024), scientific reasoning (Wang et al., 2022b) and Embodied Artificial Intelligence(Embodied AI) (Puig et al., 2018; Shridhar et al., 2020; Ma et al., 2024). These tasks often require agents to plan and execute long sequences of actions while interacting with an environment (Li et al., 2022b; Xiong et al., 2024; Li et al., 2024b; Yang et al., 2025).

Despite their powerful reasoning capabilities, LLM agents often falter in complex and unfamiliar environments due to compounding errors (Wang et al., 2024a; Xie et al., 2024b). Prevailing strategies attempt to mitigate this through post-hoc learning, using methods like environmental data exploration (Xiang et al., 2023; Xie et al., 2024b; Song et al., 2024) or refinement learning (Xiong et al., 2024; Yuan et al., 2025; Wang et al., 2025). However, these approaches share a fundamental limitation. By learning primarily from the final outcomes of entire trajectories, they struggle to pinpoint the specific, step-level actions that lead to failure. A reward for a trajectory provides a much weaker learning signal than feedback explaining why a particular action was incorrect (e.g., targeting a non-existent object or performing an invalid sequence). Without this granular, causal feedback, agents learn inefficiently from sparse signals and are prone to repeating similar mistakes.

To address this gap, we shift the paradigm from post-hoc trajectory analysis to proactive, step-level error diagnosis. We draw inspiration from software debugging. When a program fails, a developer uses a debugger to trace the execution, inspect the context, and pinpoint the exact line of code that caused the error. This pre-action, fine-grained analysis is far more effective than simply observing that the program crashed. This inspires our framework, Trajectory Graph Copilot, which incorporates a graph-based diagnostic module, Gebugger. It acts as a "debugger" for the LLM

agent, analyzing its intended action in the context of the recent past to flag potential errors before they are executed. This allows the agent to find potential errors early, preventing the accumulation of costly mistakes that would otherwise doom the entire task.

By framing agent execution within the Partially Observable Markov Decision Process (POMDP) paradigm (Xiong et al., 2024; Wang et al., 2025), our method ❶ *transforms historical trajectories into a probabilistic graph*. This structure encodes domain knowledge by capturing the underlying relationships between actions and observations (Bishop & Nasrabadi, 2006; Koller, 2009). In our formulation, nodes represent actions and edges encode observations, which can be viewed as an adaptation of the classical state–transition diagram. Unlike the traditional methods, such as the retrieve-based methods (Zhou et al., 2024), our graph-based approach offers two key advantages: it ❷ *provides higher performance* and ❸ *requires fewer samples* to achieve the same level error detection rate. We detail these insights in Section 4. Furthermore, instead of directly suggesting correct actions, GEBUGGER can ❹ *serves as a diagnosis sandbox for the LLM-Agent decision making*. By providing only potential error warnings, our method encourages LLM agents to rely on their own reasoning. This minimizes the bias introduced by fine-tuning on a fixed dataset, ultimately leading to better generalization on new tasks.

To empirically verify the effectiveness of our framework, we conduct experiments on four benchmarks, including embodied AI environments and planning tasks. We begin by collecting trajectory datasets and annotating actions with corresponding labels, followed by conducting detection experiments on them. Compared with traditional text classification and LLM-based methods, GEBUGGER demonstrates a consistent advantage in step-level error detection. Moreover, we apply GEBUGGER as a diagnosis sandbox to provide error feedback in advance. By leveraging In-Context Learning(ICL) with feedback information, our method enhances LLM agents' pass ratio by $14.69\%$ on average, outperforming baselines. In summary, the contributions are as follows:

★ We introduce, GEBUGGER, a novel method based on a probabilistic graph model, to address the challenge of step-level action error detection in LLM agents.

★ Our framework, TRAJECTORY GRAPH COPILOT, integrates this error detection module as a distinct entity. By serving as a graph-based diagnostic module for LLM agents, our approach enhances their performance by providing real-time error warnings.

★ Experiments conducted across multiple environments and LLM agents confirm the superiority of GEBUGGER in error detection and validate the effectiveness of the overall framework.

## 2 PRELIMINARIES

**Long-Horizon Tasks as POMDPs.** Following prior work, we model an LLM agent's interaction in a long-horizon task as a POMDP (Carta et al., 2023; Wang et al., 2025; Song et al., 2024; Xiong et al., 2024). A POMDP is defined by the tuple $\mathcal{M} = (\mathcal{G}, \mathcal{S}, \mathcal{A}, \mathcal{J}, \mathcal{R}, \mathcal{O}, \gamma)$, where $\mathcal{G}$ is the goal space, $\mathcal{S}$ is the state space, $\mathcal{A}$ is the action space, $\mathcal{J} : \mathcal{S} \times \mathcal{A} \to \mathcal{S}$ is the state-transition function, $\mathcal{R} : \mathcal{S} \times \mathcal{A} \times \mathcal{G} \to \mathbb{R}$ is the reward function, $\mathcal{O}$ is the observation space, and $\gamma$ is the discount factor. In the text-based environments considered by this work, the spaces $\mathcal{G}$, $\mathcal{A}$, and $\mathcal{O}$ are subsets of natural language.

**Graphical Models for Sequential Decision Making.** A Markov Decision Process can be conceptually viewed as a Probabilistic Graphical Model (PGM), where nodes represent states and edges represent transitions. This graphical perspective highlights the sequential dependencies inherent in agent trajectories. However, in a POMDP, the true state is latent, and the belief state (a probability distribution over states) is often high-dimensional and intractable to model explicitly, especially with language-based observations. Directly constructing and reasoning over a formal state-based PGM is therefore impractical. This motivates our work to develop a different, more practical graphical representation learned directly from trajectory data to diagnose action errors.

**Problem Formulation: Step-Level Error Diagnosis.** Given a particular goal $g \in \mathcal{G}$, an LLM agent generates a trajectory history of alternating observations and actions, $(o_0, a_0, \ldots, o_{t-1}, a_{t-1}, o_t)$. Before executing the next proposed action $a_t$, our goal is to determine if this action is productive or a potential error. To formalize what constitutes a "good" action, we move beyond simple binary success/failure signals. Inspired by AgentBoard (Chang et al., 2024), where the

task is decomposed into subgoals, we conceptualize complex tasks as requiring the completion of several ordered milestones. For example, in AlfWorld, the task "clean a plate and put it on countertop" involves milestones like a plate being collected, cleaned, and placed correctly. An action's quality can then be judged by its progress toward the next milestone. Figure 1 visualizes this concept, showing how an ideal "Expert Trajectory" progresses cleanly through milestone states, whereas an "Agent Trajectory" may follow a less optimal path.

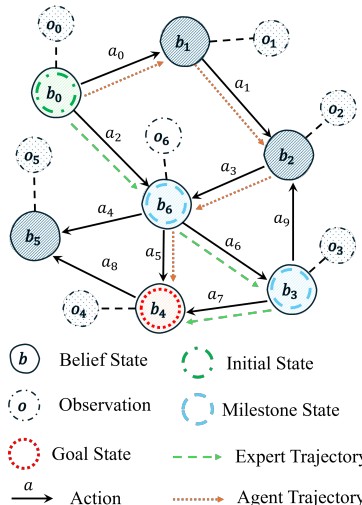

To create an informative label space $\mathcal{Y} = \{1, ..., C\}$, we define a set of fine-grained error categories that capture common failure modes in long-horizon tasks. These categories are inspired by task planning principles—using milestones to identify errors like *Precondition Not Met (P.M.)* or *Condition Met, Action Not Taken (C.M.)*—but also include general procedural errors such as *Repeated Action (R.A.)* and *Illegal Action (I.A.)*. This multi-faceted approach provides a rich, step-level diagnostic signal. The complete set of six error definitions is detailed in Appendix B.1. Our objective is to learn a diagnostic function $f$ that maps a trajectory history to a label for the *current* proposed action: $f : (o_0, a_0, \ldots, o_t, a_t) \mapsto y_t$, where $y_t \in \mathcal{Y}$. This provides granular, step-level feedback that is far more informative than a sparse, trajectory-level reward.

Figure 1: A conceptual diagram of a POMDP illustrating our error diagnosis task. An expert follows an optimal trajectory from the Initial State ($b_0$) to the Goal State ($b_4$) by progressing through Milestone States ($b_6, b_3$). In contrast, the agent's trajectory is suboptimal.

## 3 TRAJECTORY GRAPH COPILOT

In this section, we introduce a novel framework, TRAJECTORY GRAPH COPILOT for LLM agents, which includes a graph-based error diagnosis module, GEBUGGER. Initially, we transform text trajectories into graphs from the perspective of PGM. Subsequently, we adopt the idea of TextGCN (Yao et al., 2019) to regard the action error diagnosis as a node classification task. Finally, we apply GEBUGGER as a diagnosis sandbox in LLM agents for step-level action debugging and providing feedback for decision making. The overall framework is shown in Figure 2.

### 3.1 GRAPH CONSTRUCTION

Building an accurate graph based on PGM presents two challenges: representing the dependency between states and actions, and accurately representing states in the graph. From Section 2, the LLM agents' trajectories can be expressed as paths in a state transition graph. However, representing actions as edges duplicates the graph structure, reducing the overall information content, since different states may share the same action. To better extract the dependency between states and actions, a heterogeneous graph (Sutton et al., 1998), where the nodes $\mathcal{V}$ comprise both states $\mathcal{B}$ and actions $\mathcal{A}$, offers a more effective and structured representation. The heterogeneous graph enables node reduction by merging states with highly similar neighbors into a supernode, yielding a more generalized representation. However, due to the partial observation, it is hard to estimate the state in the graph. To address this issue, we consider using observations instead of states. In a POMDP, observations follow the conditional probability $p(o|s)$. Given the observation posterior probability $q(o)$ and $p(o|s)$, the state distribution can be inferred by $p(s) \propto \sum_o p(o|s)q(o)/\sum_{s'} p(o|s')$. Therefore, states are learned as implicit knowledge through observations. However, given that the observation results are in natural language and cannot be easily discretized, we integrate them as attributes within the edges to form an action-centric graph.

During implementation, to obtain a robust representation of nodes, we utilize a natural language processing tool, NLTK (Bird et al., 2009), to finish some regular preprocessing, such as removing meaningless words/numbers. We then deduplicate actions to form a set of unique nodes and construct the PGM-based graph by linking them according to their order in the trajectories. Finally, we reform the state-transition diagram as an action-centric graph $G = (\mathcal{V}, \mathcal{E})$, where $\mathcal{V} \subseteq \mathcal{A}$ and $\mathcal{E} \subseteq \mathcal{O}$.

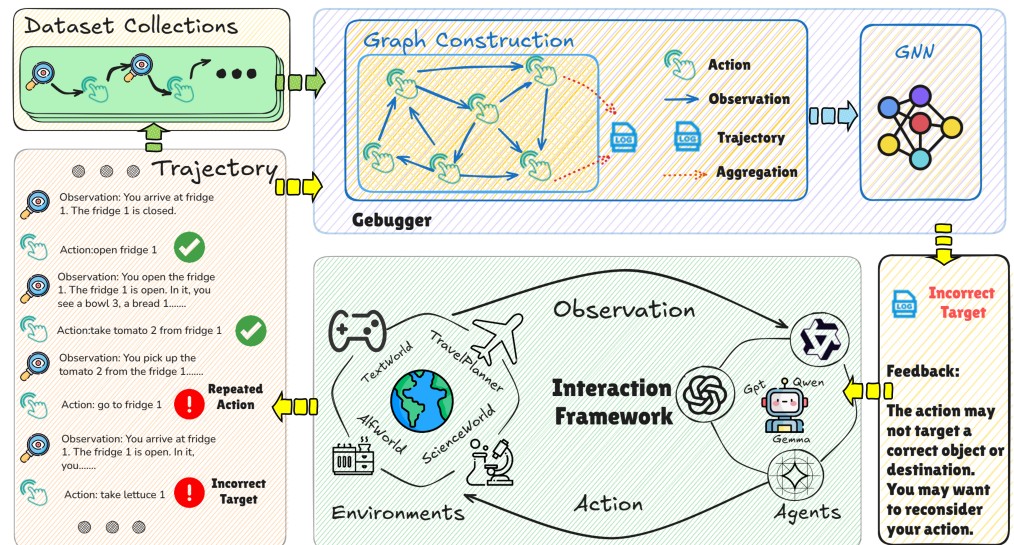

Figure 2: Overall pipeline of TRAJECTORY GRAPH COPILOT. We first collect the trajectories. We then use the GEBUGGER to predict the action error labels, which includes converting trajectories into a PGM-based graph and utilizing the GNN-based detector. Finally, our framework provides external information for agent decision-making.

## 3.2 POTENTIAL ERROR DIAGNOSIS

To achieve a step-level action error detection, we adopt the TextGCN (Yao et al., 2019) to regard trajectories as sentences. We use the pretrained Bert (Devlin et al., 2019) model to initialize action and observation embeddings. By linking the trajectory nodes with action nodes, the error probe task is converted into a trajectory node classification task. Formally, the complete graph adjacency matrix $A$ is defined as:

$$
A_{ij} = \left\{
\begin{array}{cc}
1 & v_i, v_j \in \mathcal{A}, \text{and } [v_i, o, v_j] \in \xi \\
1 & v_i \in \mathcal{A}, \text{and } v_j \in \mathcal{T} \\
0 & \text{otherwise}
\end{array}
\right. , \tag{1}
$$

where $[v_i, o, v_j]$ is a subsequence of a trajectory $\xi$, $\mathcal{T}$ is the set of trajectories and $o$ is an observation. For edge attribution, we collect all the observations $\{o, o \in [a_i, o, a_j]\}$ linking the same action pair and get the average of the Bert embedding. The rest of the nodes and edges are set up with all zeros. For potential error detection, we implement the detection mechanism outlined in the pipeline through a Graph Neural Network(GNN) architecture. Given a trajectory $\xi$, our goal is to train a mapping function $f$ that probes if there is a potential error. This corresponds to the operation of the GNN detector across $K$ rounds of message passing. Each GNN layer performs one round of message passing, defined by the following update rule:

$$
a_v^{(l)} = \text{AGG}^{(l)} \left( h_u^{(l-1)} : u \in \mathcal{N}(v) \right),
$$

$$
h_v^{(l)} = \text{COMBINE}^{(l)} \left( h_v^{(l-1)}, a_v^{(l)} \right),
$$

where $a_v^{(l)}$ denotes the aggregated message at $l$-th layer, $h_v^{(l)}$ the feature vector of node $v$, and $\mathcal{N}(v)$ its set of neighbors. The $\text{AGG}^{(l)}$ is a function that aggregates information from neighboring nodes, while $\text{COMBINE}^{(l)}$ updates the node representations, following the definition in previous work (Xu et al., 2019). After $K$ rounds of updating, this process yields the final node embeddings $\boldsymbol{H}$. For the error detection, we apply a softmax function to the final embeddings to obtain probabilities, $\boldsymbol{Z} = \text{softmax}(\boldsymbol{H})$. The GNN model is trained by minimizing the cross-entropy loss on the labeled trajectories:

$$
\mathcal{L} = - \sum_{t \in \mathcal{T}_T} \sum_{\ell \in \mathcal{Y}} Y_{t\ell} \ln Z_{t\ell}, \tag{2}
$$

where $\mathcal{T}_T$ represents the train set of trajectories and $\mathcal{Y}$ is the label space. During testing, we inject the trajectory node into the existing graph constructed by the training set. To address potential issues

where an action, observation, or task falls outside the defined scope, we use embeddings to search for and retrieve the corresponding trajectory path from the existing graph.

### 3.3 In-Context Learning Feedback

To fully leverage the corrective signals provided by our external feedback module, we design a mechanism that integrates these signals into the agent's reasoning loop through ICL (Liskavets et al., 2025; Zhang et al., 2022b; Zhou et al.). Rather than fine-tuning the underlying language model, our approach dynamically adapts the agent's behavior during inference by augmenting the prompt with structured feedback information. The detailed implementation is available in Appendix B.1.

Without feedback, the agent's generative behavior is modeled by the conditional distribution $P_{agent}(A_{i+1}|(O_0, A_0, ...O_i, A_i))$. With the environment copilot feedback, it first evaluates the $A_{i+1}$ initialized by the agent and generates the feedback signal $Y_i$, where the signal space is defined in Section 2. If $Y_i$ indicate there is potential error, then the agent should regenerate $A'_{i+1}$ according to the revised conditional distribution $P_{agent}(A'_{i+1}|(O_0, A_0, ...O_i, A_i), A_{i+1}, Y_i)$. The revised conditional distribution can be viewed as the agent's posterior over actions. In practice, ICL operationalizes this posterior update by conditioning the model on explicit feedback text in the prompt.

## 4 Theoretical Analysis

One goal of this work is to detect the potential error based on the trajectory. A straightforward approach is to directly retrieve historical trajectories to determine if the current action is erroneous. A key challenge is managing the rapidly growing size of the trajectories as the number of interactions with the environment increases. Alternatively, because the action and observation modalities are consistent, the problem of error detection can be formulated as a text sequence classification task. Prior works (Taha et al., 2024; Li et al., 2022a) have explored sequence-based modeling approaches for the task, such as Bert-based detectors (Devlin et al., 2019). Though the promise, the structured information hidden in the trajectories is not well-explored. Drawing inspiration from PGM, we formulate the action error probe task as a node classification on a converted graph. Building upon this foundation, we show that graph-based methods have a lower Bayes risk and sample complexity than sequence-based methods under the same generalization error.

To better serve theory analysis, we reorganize the trajectory as a tuple $X = (\tau, G, \boldsymbol{O}_{1:k}, \boldsymbol{A}_{1:k})$, where $G$ is the task, $\tau$ is the trajectory identifier, $O_i, A_i$ are the $i$-th observation and action for $i = 1, ...k$, and $k$ is the size of the number of action/observation in trajectory. The decision target is the ground truth class $Y \in \mathcal{Y}$ of the final action. We consider a class of baselines, the *empirical sequence-based* (ESB) method, which captures the critical characteristics of existing approaches. The classical ESB method, such as a fine-tuned Bert classification model, contains two modules, a sequence representation mapping $U = \Phi_{seq}(X) \in \mathcal{U} \subseteq \mathbb{R}^{d_{seq}}$ and a classifier $h_{seq} : \mathcal{U} \to \mathcal{Y}$. In this work, the graph representation is $S = \Phi_g(X) \in \mathcal{S} \subseteq \mathbb{R}^{d_g}$ obtained by *probabilistic graph-based* (PGB) approach, where $S$ is chosen to be the Markov blanket (Pearl, 1998) of $Y$ in the graph induced by the connectivity rules. Similarly, a classifier $h_g$ is learned from $(S, Y)$ pairs. To better compare ESB and PGB, we first make two conventional assumptions as follows.

**Assumption 4.1 (Conditional Sufficiency).** *There exists a representation $S = \mathrm{MB}_G(Y)$ (the Markov blanket of $Y$ under the graph construction) such that $Y \perp\!\!\!\perp X \setminus S \mid S$.*

This assumption indicates that the label of an action is determined in limited steps and dependent on other steps, which is aligned with the Markov decision process. In the Appendix B.1, we provide an example to illustrate why this assumption holds. In Section 2, we follow the assumption to use the milestone states to refer to the action labels.

**Assumption 4.2 (Representation Capacity Difference).** *The sequence representation $U = \Phi_{\text{seq}}(X)$ is not a deterministic function of $S$. In particular, $U$ include additional spurious components $Z$ such that $U = g(S, Z)$, with $Z$ correlated with environment-specific artifacts.*

The assumption 4.2 suggest that the ESB could capture spurious components, such as random noise, writing style. In practice, this assumption is reasonable because training a robust model to mitigate the effects of random noise requires a large dataset. Based on these assumptions, we obtain two conclusions.

Table 1: Action error detection results(%), the metric is the accuracy(↑). GPT4o. is short for GPT4o-mini. The best performances are in **bold**, and the second-best method is underlined.

| Method | AlfWorld | | | TextWorld | | | ScienceWorld | | | TravelPlanner | | |
|---|---|---|---|---|---|---|---|---|---|---|---|---|
| | GPT4o. | Qwen2.5 | Gemma3 | GPT4o. | Qwen2.5 | Gemma3 | GPT4o. | Qwen2.5 | Gemma3 | GPT4o. | Qwen2.5 | Gemma3 |
| **Text Classification** | | | | | | | | | | | | |
| TF-IDF | 58.04 | 66.98 | 63.75 | 63.68 | 49.82 | 42.09 | 81.02 | 80.66 | 80.52 | **93.35** | 65.23 | 65.22 |
| Bert | 53.09 | 62.07 | 55.85 | 64.17 | 54.12 | 49.44 | 83.51 | 80.48 | 78.72 | 92.14 | 65.76 | 64.63 |
| **Retrieve** | | | | | | | | | | | | |
| MiniLM | 39.33 | 46.74 | 49.74 | 59.37 | 43.55 | 42.37 | 73.87 | 76.57 | 73.76 | 90.20 | 60.56 | 58.79 |
| E5 | 44.82 | 57.51 | 53.14 | 57.05 | 27.24 | 52.54 | 76.59 | 74.48 | 72.98 | 90.88 | 58.22 | 59.23 |
| GTR | 48.10 | 58.41 | 54.00 | 60.20 | 30.47 | 53.67 | 76.17 | 74.74 | 74.00 | 90.09 | 59.77 | 59.30 |
| **RAG** | | | | | | | | | | | | |
| GPT4o | 55.18 | 62.73 | 60.38 | 52.74 | 49.10 | **69.21** | 72.15 | 74.21 | 68.76 | 79.87 | 56.82 | 55.72 |
| Gemma3 | 55.74 | 64.43 | 59.95 | 61.69 | 51.25 | 60.17 | 77.11 | 80.75 | 76.59 | 34.95 | 39.81 | 39.28 |
| Qwen2.5 | 50.04 | 57.56 | 55.32 | 43.12 | 35.84 | 51.69 | 67.94 | 71.05 | 66.30 | 24.25 | 31.22 | 31.29 |
| **LLM Zero-Shot** | | | | | | | | | | | | |
| GPT4o | 32.12 | 30.13 | 32.06 | 45.77 | 42.29 | 48.31 | 33.33 | 35.30 | 31.45 | 4.21 | 8.68 | 8.03 |
| Gemma3 | 31.04 | 28.92 | 33.81 | 48.26 | 34.59 | 44.35 | 18.81 | 34.42 | 20.30 | 2.42 | 4.48 | 4.03 |
| Qwen2.5 | 16.46 | 20.24 | 16.75 | 39.64 | 27.78 | 31.64 | 19.76 | 19.79 | 20.83 | 2.25 | 10.86 | 11.46 |
| **LLM One-Shot** | | | | | | | | | | | | |
| GPT4o | 22.34 | 21.81 | 28.22 | 51.41 | 37.46 | 55.65 | 13.49 | 13.43 | 11.03 | 4.32 | 8.89 | 7.96 |
| Gemma3 | 27.30 | 20.99 | 32.39 | 42.45 | 32.62 | 37.85 | 9.78 | 21.52 | 14.31 | 2.55 | 4.46 | 4.19 |
| Qwen2.5 | 17.12 | 17.59 | 17.05 | 40.30 | 33.15 | 38.98 | 31.12 | 17.96 | 21.44 | 2.58 | 11.19 | 11.55 |
| **LLM Three-Shot** | | | | | | | | | | | | |
| GPT4o | 29.13 | 20.65 | 33.51 | 50.41 | 35.13 | 49.72 | 15.85 | 15.03 | 11.77 | 3.46 | 2.48 | 4.42 |
| Gemma3 | 30.21 | 26.29 | 27.26 | 38.47 | 34.77 | 33.33 | 29.23 | 50.91 | 20.83 | 2.27 | 5.59 | 8.73 |
| Qwen2.5 | 16.33 | 19.29 | 16.59 | 44.28 | 26.70 | 37.85 | 28.54 | 31.97 | 22.63 | 0.95 | 5.50 | 5.57 |
| Ours | **63.98** | **69.89** | **66.85** | **67.00** | **62.19** | 66.67 | **87.84** | **84.57** | **83.39** | 92.75 | **67.43** | **68.15** |

**Theorem 4.3** (**Bayes Risk Ordering and Sample Complexity**). *For any measurable classifier $h$ and its Bayes risk defined by $R(h \circ \Phi) := \mathbb{P}\big(h(\Phi(X)) \neq Y\big)$, the minimal achievable risk using representation $S$ equals the Bayes risk using the full input $X$. Moreover, for any other representation $U = \Phi(X)$,*

$$R^*(S) \leq R^*(U),$$

*where $R^*(\cdot)$ denotes the Bayes (irreducible) risk for classifiers that see only that representation. If $I(Y;S) > I(Y;U)$, then the inequality is strict. Moreover, the sample complexity required to achieve classification error $\epsilon$ satisfies $m_\text{g} < m_\text{seq}$, under the same error tolerance $\epsilon$.*

The theorem demonstrates that the graph-based method has a lower Bayes risk and could achieve superior performance compared to the ESB method under identical conditions. The detailed proof is provided in Appendix A.

## 5 RELATED WORK

**LLM Agents.** Empowered by LLMs, agents have experienced rapid growth and demonstrated remarkable performance across a wide range of tasks, including goal reasoning and action execution (Xi et al., 2025).For instance, LLMs have empowered embodied agents (Chen et al., 2023b) with perception, interaction, and planning skills for versatile operation in virtual and physical environments (Londoño et al., 2024). To address the long-horizon interaction tasks, existing methods can be divided into two categories: fine-tuning-based and fine-tuning-free methods. To obtain a refined language model agent, fine-tuning-based methods (Wang et al., 2025; Xiong et al., 2024; Wang et al., 2024a; Song et al., 2024; Wang et al., 2023) enhance decision-making capabilities by tuning LLMs from expert demonstrations or exploration (Chen et al., 2023a; Yin et al., 2023; Xiang et al., 2023; Song et al., 2024). Another line of work incorporates external tools/models to gain improvements, such as structure search (Yao et al., 2023a; Besta et al., 2024; Hao et al., 2023; Zhuang et al., 2023) and retrieval (Xiao et al., 2023; Kagaya et al., 2024; Zhou et al., 2024). These methods typically guide LLM agents by incorporating external knowledge. For instance, structure search offers feedback from the environment (Xiang et al., 2023), while retrieval selects optimal actions by comparing them to offline successful trajectories (Kagaya et al., 2024).

**Graph in LLM Reasoning.** To enhance the reasoning capacity of LLM, recent works (Yao et al., 2023a; Besta et al., 2024; Wang et al., 2022c), such as chain-of-thoughts (Wei et al., 2022), are proposed. These methods essentially decompose the LLM's reasoning into nodes and edges, where nodes represent entities and edges represent thought processes (Besta et al., 2024), thereby modeling

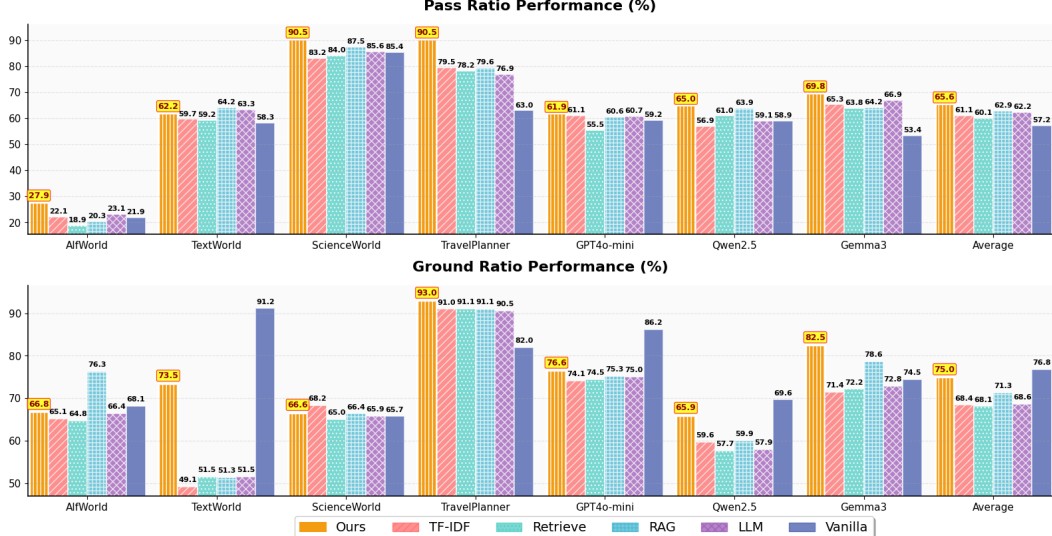

Figure 3: The pass ratio(%) and ground ratio(%) results on four benchmarks and three LLM agents.

interactive relationships. Leveraging these relationships can enhance the LLM's reasoning capabilities. Alternatively, explicit graph structures like knowledge graphs (Mavromatis & Karypis, 2025) can serve as external knowledge bases to guide reasoning, such as GraphRAG (Peng et al., 2024; Luo et al., 2024; Li et al., 2024a), Knowledge Graph Question Answering(KGQA) (Lan et al., 2022; Ye et al., 2021; Zhang et al., 2022a). In (Wang et al., 2024b), the authors propose that the reasoning ability of the language model can be seen as an aggregation of the numerous indirect 'reasoning paths' encountered during pretraining. Moreover, integrating graph structures with LLMs can enhance their reasoning abilities. For instance, (Li et al., 2025) embeds knowledge graph representations directly into LLM tokens as 'graph semantics,' enabling the model to incorporate structural information without relying on prompt engineering or extensive fine-tuning. Furthermore, graphs are increasingly used to ground LLM reasoning in actionable plans and environments. For instance, graphs can structure high-level subgoals (Lee et al., 2022), represent physical scenes for failure recovery (Yu et al., 2025), or facilitate graph-based learning to improve the planning robustness of LLM agents (Wu et al., 2024b).

## 6 EXPERIMENTS

To verify the effectiveness of our framework, we conduct empirical studies on two perspectives: error detection and feedback evaluation. We first collect and build datasets on four environments, and then we compare the performance on our framework and baselines.

**Datasets.** Four environments are used for dataset construction, including AlfWorld (Shridhar et al., 2020), TextWorld (Côté et al., 2018; Jansen & Côté, 2022), ScienceWorld (Wang et al., 2022b), and TravelPlanner (Xie et al., 2024a). These benchmarks evaluate long-term reasoning by requiring agents to interact with the environment, gather information, and make decisions. For instance, ScienceWorld evaluates an agent's ability to programmatically solve problems using scientific knowledge, while TravelPlanner tests its capacity to use tools for information gathering and planning under user constraints. We follow the React (Yao et al., 2023b) to implement an agent for the data collection and experiments. In AlfWorld, we follow Agentboard to split the train, validation, and test tasks. In TextWorld and ScienceWorld, we use the first 10 variants as train and validation tasks, and 5 variants as test tasks. In TravelPlanner, we select 100 tasks for each difficult level as test tasks and reserve the others for train and validation tasks. To remove the bias of LLMs, we use GPT4o-mini (OpenAI, 2024), Qwen2.5-14b (Yang et al., 2024), and Gemma3-27b (Team et al., 2025) for all four environments. For the step-level annotation, we first employ LLM models to generate the label, then select and filter manually. The detailed dataset information is available in Appendix B.2.

**Evaluation Metrics.** For error detection, we use the classical metric, accuracy, to measure the performance. In the feedback evaluation, we report two metrics: Pass Ratio(PR) and Ground Ratio(GR). PR is used to evaluate whether an agent has successfully completed a given task, and GR

is a metric that assesses the validity of an agent's action within a given environment state, serving as an indicator of its grounding and understanding. In TravelPlanner, the PR indicates whether the agents give the final plan. In practice, we are primarily concerned with the PR.

**Baselines.** Our baselines contain three categories of approaches. The conventional methods include TF-IDF (Salton & Buckley, 1988) and fine-tuned Bert methods (Devlin et al., 2019), which are represent the text classification approaches. We use TF-IDF to extract the feature and logistic regression to predict the error probability. We also use retrieval-based (Guu et al., 2020) and Retrieval-Augmented Generation(RAG) (Lewis et al., 2020) methods as representative approaches for incorporating external databases. In retrieval-based methods, we use embedding to find the most similar trajectory. We use three language models to obtain the embeddings, including ALL-MiniLM-L6-v2(MiniLM) (Wang et al., 2021), E5-Large(E5) (Wang et al., 2022a), and GTR-T5-Large(GTR) (Ni et al., 2022). For RAG, we select the five most similar trajectories as candidates and use LLMs to generate the answer. We use the GTR as the embedding model to measure the similarity. Furthermore, we consider the LLM-as-a-judge (Zheng et al., 2023) methods as the LLM-based methods, including three settings: zero-shot, one-shot, and three-shot. To avoid the model bias, we use three LLM models for RAG and LLM-as-judge: GPT4o (OpenAI et al., 2024), Qwen2.5-14b, and Gemma3-27b.

## 6.1 DETECTION RESULTS

In Table 1, we report the results across four benchmark datasets. Overall, our method consistently outperforms all baselines. Specifically, the graph-based detection approach achieves over a 5% improvement compared to text classification methods on average. The advantage is most striking in the TextWorld environment, where the agent is built with Gemma3, and our method achieves a 34.85% increase over the second-best approach. This observation aligns with our analysis that graph-based methods require lower data complexity than sequence-based approaches, making them more effective in handling sparse or noisy trajectories.

We also observe interesting differences across RAG methods. In the first three environments, RAG methods outperform standard retrieval approaches, whereas in TravelPlanner the opposite holds. We attribute this reversal to the much longer observation sequences in TravelPlanner, which may dilute the benefits of RAG and make simple retrieval more effective. Meanwhile, LLM-as-judge methods consistently perform poorly.

Table 2: Graph Ablation Results. The Dir. and Undir. are short for directed graph and undirected graph.

| Graph | AlfWorld | | | ScienceWorld | | | |
|---|---|---|---|---|---|---|---|
| | GPT4o. | Qwen2.5 | Gemma3 | GPT4o. | Qwen2.5 | Gemma3 | Avg. |
| **Onehot** | | | | | | | |
| Dir. | **64.22** | 69.38 | 65.60 | 85.95 | 81.99 | 81.02 | 74.69 |
| Undir. | 62.44 | 67.83 | 65.63 | 82.21 | 82.17 | 81.71 | 73.67 |
| **Bert** | | | | | | | |
| Dir. | 63.98 | 69.89 | 66.85 | **87.84** | 84.57 | 83.39 | **76.09** |
| Undir. | 63.05 | **70.18** | **67.22** | 84.32 | **85.68** | **83.72** | 75.70 |

We believe this is due to the inherent difficulty of the tasks, which demand strong logical reasoning skills beyond the capabilities of current judgment-style approaches.Finally, we find no consistent trend between one-shot and three-shot settings. Surprisingly, in most cases, the zero-shot setting achieves better performance than either, likely because additional samples introduce bias that hinders the reasoning ability of the LLMs. These findings further highlight the robustness of our graph-based detection approach across different environments and prompting strategies.

## 6.2 FEEDBACK EVALUATION RESULTS

To evaluate the effectiveness of TRAJECTORY GRAPH COPILOT, we conduct feedback evaluation experiments. We compare against four baselines: TF-IDF, GTR, Gemma3, and GPT4o, representing text-classification, retrieval-based, RAG-based, and LLM-as-judge (zero-shot) categories, respectively. The detailed experimental setup is provided in Appendix B.4. We report the average results on four benchmarks and three LLM agents in Figure 3. As the results show, both our method and baselines outperform the vanilla results in most cases, demonstrating that the mechanism, feedback in the interaction, consistently enhances agent performance across different detection strategies. Among these methods, TRAJECTORY GRAPH COPILOT achieves the strongest performance on average, which outperforms baselines on all four benchmarks. On average, our method improves the vanilla by 14.69%. Interestingly, even for RAG and LLM-as-judge, which show weaker detection

accuracy, the agents still benefit from performance gains. For example, in ScienceWorld, LLM-as-Judge methods have a poor detection ratio but still boost the agents. A possible explanation is that LLMs may not always follow instructions precisely, leading to incorrect prediction labels; however, they still provide useful analysis that guides the agents. Another notable observation is that the GR does not directly correlate with the PR. For instance, in TextWorld, though the GR of all the methods is lower than vanilla, PR ratios are still improves. This is expected, since the feedback mechanism does not modify the LLMs directly but instead encourages them to generate diverse candidate actions, even when some contain errors. For the detailed results, we provide in Appendix C.2.

## 6.3 ABLATION STUDY

To examine the graph's variance, we conducted ablation studies on another three settings, exploring combinations of two node features, Bert embedding or one-hot embedding as attribution, and two edge types, directed and undirected edges. As shown in Table 2, our results indicate that the performance of undirected graphs is slightly inferior to that of directed graphs. The key reason is that POMDP dependencies are inherently directional, and representing them with undirected graphs introduces noise, leading to a slight performance decline. Additionally, using one-hot encoding as a feature yields poorer performance compared to using BERT, demonstrating that text information is useful in the detection tasks.

Furthermore, we conduct ablation studies to analyze the impact of different settings of TRAJECTORY GRAPH COPILOT on the TextWorld and ScienceWorld benchmarks using Qwen2.5 and Gemma3. Specifically, we analyze two key factors: the maximum number of attempts and the choice of confidence threshold, as detailed in Appendix B.4. As illustrated in Figure 4, PR performance improves as the number of attempts increases, before eventually stabilizing.

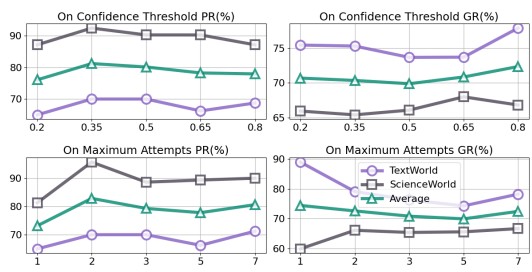

This behavior is intuitive; additional attempts raise the likelihood of producing valid and reasonable actions, thereby reducing the probability of failure at each step. However, we also observe a decline in GR, consistent with the findings in Section 6.2. In contrast, the confidence threshold has only a marginal effect on overall performance. We attribute this to the strength of the detection module: when detection is highly accurate and robust, the threshold plays a minor role, as relatively few false alarms or misclassifications propagate to the next stage. If detection were less reliable, the choice of threshold

Figure 4: The ablation study on maximum attempt times and confidence threshold.

would likely have a much greater impact. Interestingly, we also find that setting a higher confidence threshold slightly improves GR performance. Additional quantitative results supporting these observations are provided in Appendix C.3.

## 7 CONCLUSION

In this paper, we propose GEBUGGER, a novel PGM-based graph detection method for step-level diagnosis of agent action decisions. Unlike traditional approaches and LLM-based methods such as text classification, RAG, or LLM-as-judge, GEBUGGER achieves lower error rates while requiring fewer samples. Beyond the detection module itself, we further introduce TRAJECTORY GRAPH COPILOT, a flexible pipeline that integrates the detection module as an independent sandbox to provide actionable feedback on agent behaviors. We conduct extensive experiments on four benchmarks and three LLM-based agents to validate the effectiveness of our approach. For action detection, GEBUGGER consistently outperforms all baseline methods, demonstrating its robustness and efficiency. For the feedback pipeline, we observe that incorporating any detection module, whether a baseline or GEBUGGER, can enhance agent performance, but GEBUGGER provides the largest and most consistent improvements. These findings highlight not only the effectiveness of GEBUGGER in detecting errors but also its potential as a general framework for improving decision-making in LLM agents. We believe this work can inspire more reliable performance-enhancement strategies for agent-based systems.

## ETHIC STATEMENT

All authors confirm that they have read and commit to upholding the ICLR Code of Ethics. All experiments use publicly available benchmarks; no human subjects or sensitive data are involved.

## REPRODUCIBILITY STATEMENT

We will release (upon publication) all code, configuration files, and scripts needed. All experiments use publicly available benchmarks, and are reproducible.

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

CONTENTS

## A    Proofs of Theorem 4.3

*Proof.* We prove the proofs of Theorem 4.3 in several steps.

**(1) Sufficiency.**    Assumption 4.1 states exactly that $S$ d-separates $Y$ from the rest of the observed variables, i.e. $Y \perp\!\!\!\perp X \setminus S \mid S$. This is equivalent to the conditional probability factorization $\mathbb{P}(Y \mid X) = \mathbb{P}(Y \mid S)$. Hence $S$ is a sufficient statistic for $Y$ relative to $X$.

**(2) Bayes risk equality.**    Given the full input $X$, the Bayes classifier and its risk are:

$$h_X^*(x) = \arg \max_{c \in \{1,\dots,C\}} \mathbb{P}(Y = c \mid X = x), \text{and } R^*(X) = \mathbb{E}\big[1\{h_X^*(X) \neq Y\}\big].$$

Because $\mathbb{P}(Y \mid X) = \mathbb{P}(Y \mid S)$, the Bayes posterior and classifier can be written using $S$ only:

$$h_X^*(x) = \arg \max_c \mathbb{P}(Y = c \mid S = \Phi_{\mathrm{g}}(x)) \equiv h_S^*(\Phi_{\mathrm{g}}(x)).$$

Therefore $R^*(X) = R^*(S)$: no additional reduction in Bayes risk is possible by observing $X$ instead of $S$.

**(3) Information-theoretic ordering.**    For any representation $U = \Phi(X)$, consider the Markov chain $Y \rightarrow X \rightarrow U$. By the data processing inequality, $I(Y; U) \leq I(Y; X)$. Because $S$ is a (deterministic) function of $X$ and suffices for $Y$, we also have $I(Y; S) = I(Y; X)$. Hence $I(Y; U) \leq I(Y; S)$.

A standard information-theoretic lower bound relates classification error to the remaining uncertainty $H(Y) - I(Y; U)$. Therefore a representation with strictly larger mutual information with $Y$ yields a strictly smaller lower bound on achievable error; hence if $I(Y; S) > I(Y; U)$ then $R^*(S) < R^*(U)$.

**(4) Information-Theoretic Sample Complexity.**    By Fano's inequality for $C$-class classification, the minimal number of samples $m$ required to achieve error $\epsilon$ satisfies

$$m \gtrsim \frac{(1 - \epsilon) \log C - I(Y; X)}{\log C},$$

where $X$ is the embedding. Substituting the assumption in Section 4 , we obtain

$$m_{\mathrm{g}} \propto \frac{\log C - I(Y; S)}{\log C}, \quad m_{\mathrm{seq}} \propto \frac{\log C - I(Y; U)}{\log C}.$$

Since $I(Y; S) \geq I(Y; U)$, it follows that $m_{\mathrm{g}} \leq m_{\mathrm{seq}}$, with strict inequality if $I(Y; S) < I(Y; U)$. This completes the proof. $\square$

## B    Detailed Implementation

### B.1    Error Definition

As described in Section 2, we utilize the milestone state definition to detect the errors, which is based on the previous actions and goals. Besides, we also consider the repeated action as an error. In this work, we define the error using the following categories:

- No Error(N.E.): The current action towards the next milestone state.
- Illegal Action(I.A.): The current action is not a valid action for the current environment.
- Repeated Action(R.A.): The current action has been done in the trajectory, and the results are the same.
- Incorrect Target(I.T.): The current action causes the agent to grab a wrong object or move to a wrong destination.
- Precondition Not Met(P.M.): The current action is valid, but it can only be executed when the agent has finished a specific action.

Table 3: Action error statistic information of datasets.

| | Train and Validate | | | | | | Test | | | | | |
|---|---|---|---|---|---|---|---|---|---|---|---|---|
| Agent | N.E. | I.A. | I.T. | R.A. | P.M. | C.M. | N.E. | I.A. | I.T. | R.A. | P.M. | C.M. |
| **AlfWorld** | | | | | | | | | | | | |
| GPT4o-mini | 746 | 4410 | 410 | 51 | 403 | 2187 | 245 | 1546 | 126 | 14 | 179 | 771 |
| Qwen2.5 | 804 | 5612 | 544 | 552 | 2825 | 1105 | 268 | 1838 | 171 | 173 | 1082 | |
| Gemma3 | 804 | 2186 | 583 | 237 | 3334 | 1120 | 268 | 932 | 213 | 96 | 1150 | 367 |
| **TextWorld** | | | | | | | | | | | | |
| GPT4o-mini | 240 | 181 | 81 | 70 | 549 | 63 | 80 | 76 | 35 | 32 | 343 | 37 |
| Qwen2.5 | 240 | 298 | 67 | 88 | 214 | 60 | 80 | 218 | 46 | 42 | 142 | 30 |
| Gemma3 | 240 | 146 | 57 | 50 | 244 | 49 | 80 | 26 | 43 | 23 | 151 | 31 |
| **ScienceWorld** | | | | | | | | | | | | |
| GPT4o-mini | 601 | 4943 | 57 | 72 | 369 | 186 | 288 | 2924 | 13 | 21 | 306 | 57 |
| Qwen2.5 | 600 | 4332 | 23 | 59 | 383 | 54 | 288 | 1778 | 12 | 11 | 149 | 11 |
| Gemma3 | 600 | 3399 | 37 | 263 | 65 | 60 | 288 | 1953 | 8 | 127 | 21 | 42 |
| **TravelPlanner** | | | | | | | | | | | | |
| GPT4o-mini | 12441 | 507 | 41 | 394 | 41 | 12 | 4210 | 177 | 24 | 127 | 3 | 0 |
| Qwen2.5 | 9362 | 3731 | 143 | 502 | 37 | 18 | 2826 | 1181 | 57 | 220 | 14 | 10 |
| Gemma3 | 9341 | 3734 | 140 | 547 | 35 | 20 | 2821 | 1174 | 61 | 216 | 13 | 10 |

- Condition Met, Action Not Taken(C.M.): The current action is a valid action, but not necessary for the next milestone state or goal.

Notably, these error definitions are not specific to any particular environment. As a result, they do not include environment-related errors, such as tool misuse, which makes them easy to extend to other environments.

To further illustrate the error space, we provide an example. Given a task such as "clean a plate and put it on the countertop", the key states include: (1) a plate is found, (2) the plate is cleaned, and (3) the plate is placed on the countertop. Suppose the action-only trajectory is "go to desk, go to basin, pick up plate 1, clean plate 1, go to desk 1." In this trajectory, the label for "go to desk 1" is "Condition Met, Action Not Taken." Making this prediction only requires tracing back to the action "clean plate 1", because after that action the plate becomes clean. This example demonstrates how the error space is defined and why the Markov blanket assumption is reasonable.

Table 4: Action and trajectory statistic information of datasets.

| | Trajectory | | | Action | | |
|---|---|---|---|---|---|---|
| Agent | train | validate | test | train | validate | test |
| **AlfWorld** | | | | | | |
| GPT4o-mini | 7386 | 821 | 2881 | 74127 | 8286 | 28704 |
| Qwen2.5 | 10297 | 1145 | 3883 | 63522 | 7288 | 25020 |
| Gemma3 | 7437 | 827 | 3026 | 41432 | 4817 | 16352 |
| **TextWorld** | | | | | | |
| GPT4o-mini | 1065 | 119 | 603 | 9760 | 1057 | 4706 |
| Qwen2.5 | 870 | 97 | 558 | 10286 | 1161 | 5923 |
| Gemma3 | 707 | 79 | 354 | 7131 | 721 | 4174 |
| **ScienceWorld** | | | | | | |
| GPT4o-mini | 5605 | 623 | 3609 | 30415 | 3506 | 16497 |
| Qwen2.5 | 4905 | 546 | 2249 | 17557 | 1818 | 5277 |
| Gemma3 | 3981 | 443 | 2439 | 17713 | 1953 | 7659 |
| **TravelPlanner** | | | | | | |
| GPT4o-mini | 12055 | 1340 | 4541 | 82691 | 9201 | 31017 |
| Qwen2.5 | 12413 | 1380 | 4308 | 72468 | 7996 | 23604 |
| Gemma3 | 12435 | 1382 | 4295 | 72260 | 8079 | 23547 |

## B.2 DATASET CONSTRUCTION

To build the datasets, we use three LLMs for each environment. For AlfWorld, TextWorld, and ScienceWorld, we use GPT4o-mini (OpenAI, 2024), Qwen2.5-14b (Yang et al., 2024), and Gemma3-27b (Team et al., 2025) to construct the Agent. We first run the training tasks to collect the agent trajectories and obtain the expert/golden trajectory from environments. We then utilize an LLM to

Table 5: Extended action error detection results(%), the metric is the average accuracy(↑). GPT4o. is short for GPT4o-mini. The best performances are in **bold**, and the second-best method is underlined.

| Method | AlfWorld | TextWorld | ScienceWorld | TravelPlanner | GPT4o. | Qwen2.5 | Gemma3 |
|--------|----------|-----------|--------------|---------------|--------|---------|--------|
| **Text Classification** | | | | | | | |
| TF-IDF | 62.92 | 51.86 | 80.73 | 74.60 | 74.02 | 65.67 | 62.90 |
| Bert | 57.00 | 55.91 | 80.90 | 74.18 | 73.23 | 65.61 | 62.16 |
| **Retrieve** | | | | | | | |
| MiniLM | 45.27 | 48.43 | 74.73 | 69.85 | 65.69 | 56.86 | 56.17 |
| E5 | 51.82 | 45.61 | 74.68 | 69.44 | 67.34 | 54.36 | 59.47 |
| GTR | 53.50 | 48.11 | 74.97 | 69.72 | 68.64 | 55.85 | 60.24 |
| **RAG** | | | | | | | |
| GPT4o | 59.43 | 57.02 | 71.71 | 64.14 | 64.99 | 60.72 | 63.52 |
| Gemma3 | 60.04 | 57.70 | 78.15 | 38.01 | 57.37 | 59.06 | 59.00 |
| Qwen2.5 | 54.31 | 43.55 | 68.43 | 28.92 | 46.34 | 48.92 | 51.15 |
| **LLM Zero-Shot** | | | | | | | |
| GPT4o | 31.44 | 45.46 | 33.36 | 6.97 | 28.86 | 29.10 | 29.96 |
| Gemma3 | 31.26 | 42.40 | 24.51 | 3.64 | 25.13 | 25.60 | 25.62 |
| Qwen2.5 | 17.82 | 33.02 | 20.13 | 8.19 | 19.53 | 19.67 | 20.17 |
| **LLM One-Shot** | | | | | | | |
| GPT4o | 24.12 | 48.17 | 12.65 | 7.06 | 22.89 | 20.40 | 25.72 |
| Gemma3 | 26.89 | 37.64 | 15.20 | 3.73 | 20.52 | 19.90 | 22.19 |
| Qwen2.5 | 17.25 | 37.48 | 23.51 | 8.44 | 22.78 | 19.97 | 22.26 |
| **LLM Three-Shot** | | | | | | | |
| GPT4o | 27.76 | 45.09 | 14.22 | 3.45 | 24.71 | 18.32 | 24.86 |
| Gemma3 | 27.92 | 35.52 | 33.66 | 5.53 | 25.05 | 29.39 | 22.54 |
| Qwen2.5 | 17.40 | 36.28 | 27.71 | 4.01 | 22.53 | 20.87 | 20.66 |
| Ours | **66.91** | **65.29** | **85.27** | **76.11** | **77.89** | **71.02** | **71.27** |

analyze the trajectory to generate initial labels. Finally, we manually review and correct these labels. The Gemma3-27b was used for the initialization labels. For the detailed prompts, we provide them at the end of the Appendix. Then we filter and map the label manually. For AlfWorld, we use the AgentBoard dataset, with 402 tasks for training and validation, and 134 tasks for testing. The TextWorld environment comprises eight subsets; for each, we use 10 variants for training/validation and 5 variants for testing. Similarly, ScienceWorld has 30 subsets, with the first 10 variants used for training/validation and the subsequent 5 variants for testing. For all three of these environments, expert trajectories serve as "No Error" samples. The TravelPlanner dataset contains three levels of tasks(easy, normal, and hard), with 880 tasks for training/validation and 300 tasks for testing (100 for each level). The detailed statistics information is available in Table 4 and 3. All data and code will be released once the paper is accepted.

Table 6: Statistical significance analysis of action error detection on TextWorld.The best performances are in **bold**, and the second-best method is underlined.

| Method | Accuracy/Micro F1 | | | Macro F1 | | |
|--------|-------------------|------|------|----------|------|------|
| | GPT4o. | Qwen2.5 | Gemma3 | GPT4o. | Qwen2.5 | Gemma3 |
| **Text Classification** | | | | | | |
| TF-IDF | $0.6365_{\pm 0.0016}$ | $0.4996_{\pm 0.0041}$ | $0.4045_{\pm 0.0160}$ | $0.3399_{\pm 0.0093}$ | $0.2712_{\pm 0.0042}$ | $0.2256_{\pm 0.0074}$ |
| Bert | $0.6368_{\pm 0.0097}$ | $0.4996_{\pm 0.0501}$ | $0.4345_{\pm 0.0493}$ | $0.4177_{\pm 0.0120}$ | $0.3897_{\pm 0.0380}$ | $0.3711_{\pm 0.0283}$ |
| **Retrieve** | | | | | | |
| GTR | $0.5322_{\pm 0.0073}$ | $0.3018_{\pm 0.0009}$ | $0.5322_{\pm 0.0073}$ | **$0.4257_{\pm 0.0041}$** | $0.2472_{\pm 0.0056}$ | $0.4257_{\pm 0.0041}$ |
| **RAG** | | | | | | |
| Gemma3 | $0.6232_{\pm 0.0030}$ | $0.5183_{\pm 0.0066}$ | $0.6124_{\pm 0.0072}$ | $0.4217_{\pm 0.0052}$ | **$0.4649_{\pm 0.0036}$** | $0.4952_{\pm 0.0052}$ |
| **LLM Zero-Shot** | | | | | | |
| Gemma3 | $0.4732_{\pm 0.0030}$ | $0.3427_{\pm 0.0029}$ | $0.4492_{\pm 0.0000}$ | $0.2522_{\pm 0.0000}$ | $0.2462_{\pm 0.0009}$ | $0.2680_{\pm 0.0000}$ |
| **LLM One-Shot** | | | | | | |
| Gemma3 | $0.3755_{\pm 0.0755}$ | $0.2620_{\pm 0.0741}$ | $0.3090_{\pm 0.0554}$ | $0.2157_{\pm 0.0408}$ | $0.1824_{\pm 0.0606}$ | $0.1799_{\pm 0.0442}$ |
| **LLM Three-Shot** | | | | | | |
| Gemma3 | $0.1158_{\pm 0.0457}$ | $0.3194_{\pm 0.0505}$ | $0.1876_{\pm 0.0907}$ | $0.0994_{\pm 0.0441}$ | $0.1932_{\pm 0.0612}$ | $0.1474_{\pm 0.0694}$ |
| Ours | **$0.6872_{\pm 0.0091}$** | **$0.5810_{\pm 0.0133}$** | **$0.6316_{\pm 0.0229}$** | $0.4226_{\pm 0.0160}$ | $0.4298_{\pm 0.0063}$ | **$0.5283_{\pm 0.0199}$** |

Table 7: Action error feedback evaluation results. The metric is PR(↑). GPT4o. is short for GPT4o-mini. The best performances are in **bold**, and the second-best method is underlined.

| Method | AlfWorld | | | TextWorld | | | ScienceWorld | | | TravelPlanner | | |
|---|---|---|---|---|---|---|---|---|---|---|---|---|
| | GPT4o. | Qwen2.5 | Gemma3 | GPT4o. | Qwen2.5 | Gemma3 | GPT4o. | Qwen2.5 | Gemma3 | GPT4o. | Qwen2.5 | Gemma3 |
| Vanilla | 18.66 | 40.30 | 6.72 | 45.00 | 65.00 | 65.00 | 77.26 | 75.69 | 86.81 | 79.33 | 54.67 | 55.00 |
| **Text Classification** | | | | | | | | | | | | |
| +TF-IDF | 22.22 | 31.34 | 12.69 | **65.00** | 57.50 | 65.00 | 71.03 | 77.93 | 86.39 | 80.67 | 60.67 | **97.00** |
| **Retrieve** | | | | | | | | | | | | |
| +GTR | 15.67 | 32.09 | 8.96 | 37.50 | **72.50** | 67.50 | 67.32 | 79.86 | 83.33 | 80.00 | 59.33 | 95.33 |
| **RAG** | | | | | | | | | | | | |
| +Gemma3 | 20.71 | 37.31 | 2.99 | 50.00 | **72.50** | 70.00 | 70.08 | **86.11** | 87.50 | **82.67** | 59.67 | 96.33 |
| **LLM Zero-Shot** | | | | | | | | | | | | |
| +GPT4o | 18.66 | 33.58 | **17.16** | 52.50 | 65.00 | **72.50** | 71.35 | 80.41 | 85.42 | 80.67 | 57.33 | 92.67 |
| +Ours | **26.12** | **44.78** | 12.69 | 46.51 | 65.00 | **75.00** | **94.44** | 82.64 | **94.44** | 80.67 | **67.67** | **97.00** |

Table 8: Action error feedback evaluation results. The metric is GR(↑). GPT4o. is short for GPT4o-mini. The best performances are in **bold**, and the second-best method is underlined.

| Method | AlfWorld | | | TextWorld | | | ScienceWorld | | | TravelPlanner | | |
|---|---|---|---|---|---|---|---|---|---|---|---|---|
| | GPT4o. | Qwen2.5 | Gemma3 | GPT4o. | Qwen2.5 | Gemma3 | GPT4o. | Qwen2.5 | Gemma3 | GPT4o. | Qwen2.5 | Gemma3 |
| Vanilla | 73.87 | **68.02** | 62.49 | **95.66** | **86.79** | **91.07** | **77.35** | 49.62 | 70.22 | 97.71 | 74.14 | 74.20 |
| **Text Classification** | | | | | | | | | | | | |
| +TF-IDF | 74.22 | 58.12 | 63.04 | 53.33 | 46.07 | 48.02 | 71.03 | **58.69** | **75.03** | 97.83 | 75.58 | 99.58 |
| **Retrieve** | | | | | | | | | | | | |
| +GTR | 74.73 | 54.89 | 64.66 | 58.11 | 43.84 | 52.58 | 67.32 | 55.42 | 72.16 | 97.73 | 76.48 | 99.21 |
| **RAG** | | | | | | | | | | | | |
| +Gemma3 | 75.78 | 61.88 | **91.19** | 57.35 | 44.24 | 52.33 | 70.08 | 57.58 | 71.41 | 97.83 | 76.04 | 99.53 |
| **LLM Zero-Shot** | | | | | | | | | | | | |
| +GPT4o | **75.84** | 57.55 | 65.88 | 55.20 | 46.57 | 52.82 | 71.35 | 52.12 | 74.12 | 97.76 | 75.36 | 98.52 |
| +Ours | 71.19 | 62.33 | 66.97 | 67.89 | 62.78 | 89.79 | 69.09 | 57.26 | 73.55 | **98.07** | **81.23** | **99.81** |

## B.3 AGENT IMPLEMENTATION

In this paper, we follow the AgentBoard (Chang et al., 2024) to build interactive agents for AlfWorld, TextWorld, and ScienceWorld. Each task is capped at 50 steps. At every step, we supply the LLM with the full history of observations and actions to support decision-making, along with a short demonstration included in the prompt to improve performance. For AlfWorld and TextWorld, we also provide the list of available action options derived from the environment. In ScienceWorld, however, the action space is prohibitively large, so we instead provide an action template. For TravelPlanner, we adopt the two-stage methods from open-source implementation[1]. The detailed prompts used in our experiments are provided at the end of the Appendix.

## B.4 TRAJECTORY GRAPH COPILOT IMPLEMENTATION

In this paper, we implement the TRAJECTORY GRAPH COPILOT using in-context learning with prompts. At each step, the agent is allowed up to three attempts to pass GEBUGGER. If a step fails, we provide additional feedback, including the error type, definition, and failure examples, to guide the regeneration of actions. In our experiments, we provide three failure samples by default. To reduce false alarms, we apply a confidence threshold of 0.6 by default to filter out unreliable detection results. For the baseline methods, we adopt the same three-attempt strategy for error detection. However, since methods such as retrieval-based approaches and LLM-as-judge do not produce confidence scores, the threshold cannot be applied. Within this feedback loop, we integrate potential error information directly into the prompt. The detailed prompts are provided at the end of the Appendix, excluding the overlapping parts already described in the agent implementation section.

## B.5 DETECTION IMPLEMENTATION

In this paper, we use a three-layer GCN (Kipf, 2016) model as the graph detector with hidden dimensions [786,512,6]. We use AdamW (Loshchilov & Hutter, 2019) as the optimizer with learning ratio 1E-3 and weight decay 5E-5. The TF-IDF method is implemented by using sklearn [2] with

---

[1] https://github.com/OSU-NLP-Group/TravelPlanner
[2] https://github.com/scikit-learn/scikit-learn

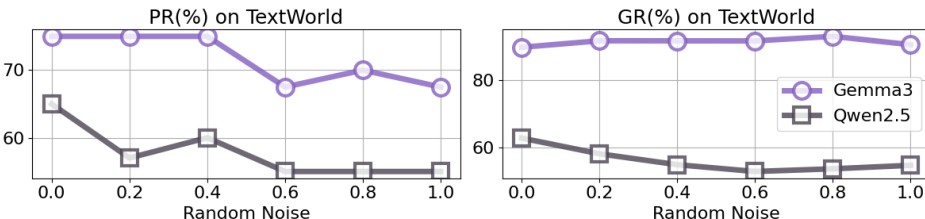

Figure 5: The pass ratio(%) and ground ratio(%) results on random noise injection.

Table 9: Statistical significance analysis of error feedback on TextWorld.The best performances are in **bold**.

| | Pass Ratio(%) | | Ground Ratio(%) | |
|---|---|---|---|---|
| Method | Qwen2.5 | Gemma3 | Qwen2.5 | Gemma3 |
| Vanilla | $65.00_{\pm6.52}$ | $70.50_{\pm5.50}$ | $\mathbf{88.77}_{\pm2.98}$ | $\mathbf{91.49}_{\pm3.42}$ |
| | **Text Classification** | | | |
| +TF-IDF | $61.50_{\pm3.35}$ | $67.50_{\pm7.07}$ | $40.49_{\pm2.35}$ | $46.81_{\pm1.51}$ |
| | **Retrieve** | | | |
| +GTR | $67.50_{\pm6.12}$ | $73.50_{\pm6.75}$ | $41.16_{\pm1.73}$ | $51.52_{\pm1.13}$ |
| | **RAG** | | | |
| +Gemma3 | $\mathbf{69.50}_{\pm3.71}$ | $71.94_{\pm5.36}$ | $40.81_{\pm0.82}$ | $52.66_{\pm1.42}$ |
| +Ours | $64.00_{\pm1.22}$ | $\mathbf{76.00}_{\pm3.74}$ | $73.61_{\pm6.74}$ | $87.15_{\pm4.78}$ |

default hyperparameters. For Bert, we finetune the model with AdamW, the learning rate 5E-5. For the retrieve and RAG methods, we use the training set as the database. For the RAG and LLM-as-Judge methods, we provide prompts at the end of the Appendix.

# C   DETAILED RESULTS

## C.1   ERROR DETECTION

In Section 6.1, we provide the detection results on four benchmarks. To better compare the performance in each benchmark and LLM, we provide the extended results. In Table 13, we observe that our method outperforms other baselines in four benchmarks and three LLM Agents. The results demonstrate the effectiveness of our method. To assess the robustness of the detection methods, we repeat the experiments five times and report the results with error bars on TextWorld. As shown in Table 9, our method consistently outperforms all baselines, aligned with our theoretical analysis.

Table 10: Comparison between ours and modern prompts on TextWorld. The best performances are in **bold**.

| | Pass Ratio(%) | | Ground Ratio(%) | |
|---|---|---|---|---|
| Method | Qwen2.5 | Gemma3 | Qwen2.5 | Gemma3 |
| ReAct | 62.50 | 70.00 | 69.86 | 86.60 |
| Reflextion | 60.00 | 67.50 | **71.18** | 79.60 |
| Ours | **65.00** | **75.00** | 62.78 | **89.78** |

Table 11: Ablation study of feedback integration mechanisms on TextWorld. In the table, "exp." is short for "explanations". The best performances are in **bold**.

| | Pass Ratio(%) | | Ground Ratio(%) | |
|---|---|---|---|---|
| Method | Qwen2.5 | Gemma3 | Qwen2.5 | Gemma3 |
| in trajectories | 57.50 | 70.00 | 52.49 | 86.69 |
| in system | 60.00 | **75.00** | 57.41 | 88.10 |
| in user | **65.00** | **75.00** | 62.78 | **89.79** |
| in user w/o exp. | 62.50 | 72.50 | **71.24** | 84.79 |

Table 12: Ablation study on confidence threshold and maximum attempts of feedback evaluation.

| | TextWorld(PR) | | ScienceWorld(PR) | | TextWorld(GR) | | ScienceWorld(GR) | |
|---|---|---|---|---|---|---|---|---|
| Hyperparameter | Qwen2.5 | Gemma3 | Qwen2.5 | Gemma3 | Qwen2.5 | Gemma3 | Qwen2.5 | Gemma3 |
| Vanilla | 65.00 | 65.00 | 75.69 | 86.81 | 86.79 | 91.07 | 49.62 | 70.22 |
| **Threshold-0.6** | | | | | | | | |
| + 1 Attempts | 65.00 | 65.00 | 75.69 | 86.81 | 86.79 | 91.07 | 49.62 | 70.22 |
| + 2 Attempts | 67.50 | 72.50 | 96.81 | 94.44 | 65.05 | 93.09 | 60.59 | 71.67 |
| + 3 Attempts | 65.00 | 75.00 | 82.64 | 94.44 | 62.78 | 89.79 | 57.26 | 73.55 |
| + 5 Attempts | 62.50 | 70.00 | 82.64 | 95.92 | 58.02 | 90.56 | 57.51 | 73.71 |
| + 7 Attempts | 70.00 | 72.50 | 86.11 | 93.75 | 64.71 | 91.71 | 59.91 | 73.47 |
| **3 Attempts** | | | | | | | | |
| + Threshold-0.20 | 62.50 | 67.50 | 77.78 | 96.53 | 60.10 | 90.80 | 56.59 | 75.32 |
| + Threshold-0.35 | 65.00 | 75.00 | 88.36 | 96.53 | 58.16 | 92.46 | 57.91 | 72.91 |
| + Threshold-0.50 | 62.50 | 77.50 | 87.50 | 93.06 | 55.16 | 92.20 | 58.82 | 73.36 |
| + Threshold-0.65 | 57.50 | 75.00 | 86.11 | 94.44 | 58.31 | 89.11 | 61.28 | 74.72 |
| + Threshold-0.80 | 65.00 | 72.50 | 88.19 | 86.11 | 63.36 | 92.43 | 58.74 | 74.91 |

Table 13: Ablation study on error samples of feedback evaluation.

| | TextWorld(PR) | | ScienceWorld(PR) | | TextWorld(GR) | | ScienceWorld(GR) | |
|---|---|---|---|---|---|---|---|---|
| Hyperparameter | Qwen2.5 | Gemma3 | Qwen2.5 | Gemma3 | Qwen2.5 | Gemma3 | Qwen2.5 | Gemma3 |
| Vanilla | 65.00 | 65.00 | 75.69 | 86.81 | 86.79 | 91.07 | 49.62 | 70.22 |
| **Zero-Shot + Threshold-0.6** | | | | | | | | |
| + 2 Attempts | 60.00 | 72.50 | 88.19 | 93.06 | 71.07 | 90.50 | 66.28 | 72.51 |
| + 3 Attempts | 50.00 | 77.50 | 87.50 | 94.44 | 69.20 | 89.28 | 63.64 | 73.86 |
| + 5 Attempts | 50.00 | 70.00 | 91.67 | 91.67 | 67.97 | 92.16 | 66.28 | 74.49 |
| **3 Attempts + Threshold-0.6** | | | | | | | | |
| + One Shot | 60.00 | 72.50 | 94.44 | 97.22 | 71.19 | 88.55 | 65.87 | 74.20 |
| + Three Shots | 65.00 | 75.00 | 82.64 | 94.44 | 62.78 | 89.79 | 57.26 | 73.55 |
| + Five Shots | 65.00 | 77.50 | 82.64 | 93.06 | 55.82 | 92.01 | 56.21 | 73.29 |

## C.2 FEEDBACK EVALUATION

In Section 6.1, we provide the average on four benchmarks and three LLM agents. In this sub-section, we provide detailed results. As PR shows in Table 7, our method achieves 7 best and 4 second-best results, demonstrating the advantage of TRAJECTORY GRAPH COPILOT. In addition, most cases outperform the vanilla method, showing that the feedback information boosts the agent's performance. In Table 8, we observe that our method achieves the second-best GR. A potential reason is that our method has a lower false alarm, resulting in less action exploration comparing to the vanilla results. To show the statistical significance analysis, we repeat the experiments for five time and report the error bars. We provide the results in Table 9.

To further discuss differences between our method and existing approaches, we compare it against ReAct and Reflexion. As shown in Table 10, our method outperforms both. This is primarily because ReAct and Reflexion rely heavily on the inherent inference capabilities of the LLM, which can lead to error accumulation and degraded performance, especially when the underlying model is not sufficiently large and no external feedback signals are available.

## C.3 ABLATION STUDY

In Section 6.3, we examine the impact of two key hyperparameters in TRAJECTORY GRAPH COPI-LOT: the maximum number of attempts and the confidence threshold. We provide the detailed results in Table 12. In this section, we further investigate the role of error samples in enhancing performance. Table 13 summarizes the results under two settings: with and without error samples. Our findings show that, without error samples, increasing the number of attempts leads to consistent improvements in PR performance, while GR remains largely unchanged. When error samples are included, PR performance also improves; however, adding more samples does not yield further gains and, in fact, leads to a decline in GR. This suggests a trade-off: while error samples provide useful information for correction, an excessive number of them may introduce noise or bias, ultimately hindering generalization.

To evaluate the effectiveness of our method, we conducted ablation experiments on the feedback component. In these experiments, we injected random noise into the detection results with noise intensities of [0.0, 0.2, 0.4, 0.6, 0.8, 1.0]. The results are shown in Figure 5. As the noise intensity increases, the model's performance gradually decreases.

We conduct an ablation study on the effects of feedback integration mechanisms. We have four settings on the format of the feedback prompt, including integrating feedback in the user prompt, feedback without label explanation, integrating feedback in the system prompt, and adding feedback in the trajectory history. We provide results in Table 11. From the results, we found Gemma3 has a robust performance on various formats while Qwen2.5 is sensitive to the format.

## C.4 PROMPTS

In this subsection, we provide the prompts used in experiments.

---

**LLM Prompt for AlfWorld, TextWorld, ScienceWorld Action Error Label**

```
"""
You are an top-level expert AI agent trajectory analyst. Your
primary task is to analyze the 'CURRENT STEP' of an agent's
trajectory and identify the single most likely error.

**Crucial Constraint**: The agent is unfamiliar with the
environment, so it might take some time to explore and find
destinations and objects. Therefore, its actions should be judged
based on the **Overall Task Goal**, not by strictly following
the expert plan. The expert plan is only a reference for a possible
successful path. You can know how much the agent has inferred the
environment from the observation and extract the best possible
action from the observation.

**Output Format**: Your entire response MUST be a single, valid JSON
object. Do not include any text, explanations, or markdown
formatting before or after the JSON object.
The JSON object must contain exactly three keys:
- "has_error": (boolean) true if there is an error(No Error is not
an error), otherwise false.
- "error_type": (string) One of the specified error types below.
- "analysis": (string) A concise, one-sentence explanation for the
error.For example, for a "Precondition Not Met" error, a good
analysis would be "The agent tried to place an object that it was
not holding in its inventory."

**Types**:
- "No Error": The action is logical and contributes to the task
goal.
- "Precondition Not Met": The action is valid but cannot be gain
progress because a necessary prior condition is not met (e.g.,
trying to 'put' an object that is not currently held).
- "Condition Met, Action Not Taken": A critical step was available,
necessary to progress and the agent already observed the condition,
but the agent performed an irrelevant or less optimal action
instead (e.g., finding the target object but not picking it up).
- "Incorrect Target": The action is performed on the wrong object or
at the wrong location (e.g., picking up a 'cloth' instead of the
required 'soapbottle').
- "Repeated Action": The action is part of a sequence that was
already performed and without any progress.

--- CONTEXT ---
1.  **Overall Task Goal**:
    "{task_goal}"
```

```
2.  **Expert's Suggested Plan (for reference only, may be flawed)**:
    {expert_plan}
3.  **Trajectory History (Recent Steps)**:
    {trajectory_history}
4.  **CURRENT STEP TO ANALYZE**:
    - Action: "{current_action}"

--- YOUR RESPONSE (JSON ONLY) ---
"""
```

LLM Prompt for TravelPlanner Action Error Label

```
prompt = f"""Please analyze the following action in a travel
planning trajectory and classify it into one of these error
categories. Before giving a conclusion, please read the Agent
thought(indicate the next step) and observation, reason the
state of the agent. In general, the action should be consistent
with the thought.

ERROR CATEGORIES:
1. No error: There is no error during the trajectory
2. Illegal Action: The action is not a valid action
3. Repeated Action: The action has been done in the trajectory, and
there is nothing updated. It is not necessary.
4. Incorrect Target: The action is not aligned with the goal, for
example the thought would go to place A, but the action searches
place B.
5. Precondition Not Met: The action is valid, but it can't be done
right now, because some condition is not met. For example, the
current information does not contain the hotel information, so it
is not a good time to make final plan. Or the current information
does not have a valid plan due to the constraints, such as money.
6. Precondition Met, Action Not Taken: The information needed is
collected. The agent can have a valid plan according to the
information, but it did not make the final plan immediately.

TASK DESCRIPTION:
{current_step.get('task_description', 'N/A')}

TRAJECTORY CONTEXT:
{context_str}

CURRENT STEP TO ANALYZE:
Agent Thought: {current_step.get('thought', 'N/A')}
Action: {current_step.get('action', 'N/A')}

Based on the context and current step, return strict JSON with
two keys: label and reason. The label must be exactly one of the
categories above (e.g., "No error", "Illegal Action", etc.). The
reason should be a brief phrase explaining why.

Attention: Only the raw json."""
```

Agent Prompt for AlfWorld environment

```
messages = [{
"role": "system",
"content": f"""You are an agent in a text-based ALFWorld environment,
performing a household task.\n
```

```
For each step, generate one action based on the task description,
action Options, current observation, and the plan.\n
Please think about the environmental state from the historical
trajectories before you give the answer.\n
Do not generate other text except the action itself.
One action per step.\n
The action output format must be ##Action: XXX ##,
where XXX is the action.\n\n
Example:\n> Observation: You are in the middle of a room.
Looking quickly around you, you see a bathtubbasin 1, a cabinet 2,
a cabinet 1, a countertop 1, a garbagecan 1, a handtowelholder 1,
a sinkbasin 1, a toilet 1, a toiletpaperhanger 1, and
a towelholder 1.\n
Your task is to: put a toiletpaper in toiletpaperhanger.\n
> Action: go to toiletpaperhanger 1\n
> Observation: On the toiletpaperhanger 1, you see nothing.\n
> Action: go to toilet 1\n> Observation: On the toilet 1, you see a
soapbottle 1, and a toiletpaper 1.\n
> Action: take toiletpaper 1 from toilet 1\n> Observation: You pick
up the toiletpaper 1 from the toilet 1.\n
> Action: go to toiletpaperhanger 1\n> Observation: On the
toiletpaperhanger 1, you see nothing.\n
> Action: put toiletpaper 1 in/on toiletpaperhanger 1\n"""},
{"role": "user",
"content": f"""
> Task Description: {task_description}
> Action Options: {','.join(action_lists)}
> Trajectory History: {' '.join(history)}
> Action:"""} ]
```

**Agent Prompt for TextWorld environment**

```
messages = [{
"role": "system",
"content": f"""You are an agent in a text-based TextWorld
environment, performing a task.\n
For each step, generate one action based on the task description,
action Options, current observation, and the plan.\n
Please think the environment state from the history trajectories
before you give the answer.\n
Do not generate other text except the action itself. One action
per step.\n
The action output format must be ##Action: XXX ##, where XXX is the
action.\n\n
Example:\n
> Observation: You are in a room. You see a coin, a table,
and a door to the north (open).\n
Your task is to: find and collect a coin.\n
> Action: look\n
> Observation: You see a coin.\n
> Action: take coin\n
> Observation: You take the coin.\n"""},
{"role": "user",
"content": f"""
> Task Description: {task_description}
> Action Options: {','.join(action_list)}
> Trajectory History: {' '.join(history)}
> Action:""" }]
```

---

**Agent Prompt for ScienceWorld environment**

```
 messages = [ {
role="system",
content=( "You are an agent in a text-based ScienceWorld
environment, performing a task.\n"
"For each step, generate one action based on the task description,
action Options, current observation, and the plan.\n"
"Please think the environment state from the history trajectories
before you give the answer.\n"
"Possible Available Actions contains the OBJ, which can be replace
by object in environment.\n"
"Do not generate other text except the action itself. One action
per step.\n"
"The action output format must be ##Action: XXX ##, where XXX is
the action.\n\n"
"Example:\n
> Observation: This room is called the hallway. In it, you see: a
picture, a substance called air, the agent.\n"
"You also see: doors to other rooms.\nYour task is to: boil water.\n
> Action: look around\n
> Observation: The door is already open.\n
> Action: open door to kitchen\n" ) },
{ role="user",
content=(
f"> Task Description: {task_description}\n"
f"> Action Options: {',',join(action_lists)}\n"
f"> Trajectory History: {' '.join(history)}\n"
f"> Action:\n" ) } ]
```

---

**LLM Prompt with the Error Feedback**

```
feedback_section = f"\n\nIMPORTANT: We have analyzed
your last time action {action} for the same trajectory and provided
this feedback: {gnn_feedback}\n Do not generate the same action,
{action}, again.\n"

messages = [{...},
{"role": "user",
"content": f"""
> Task Description: {task_description}
> Action Options: {',',join(action_lists)}
> Trajectory History: {' '.join(history)}
> Action:
{feedback_section} """}]
```

---

**LLM-as-Judge prompt for Error Detection**

```
prompt = """You are a helpful assistant for classifying
agent actions.Please give the label for the following
input. Here are labels:\n

- No Error: The action is logical and contributes to the task
goal.
- Illegal Action: The action performed an illegal action or has
no effect.
- Precondition Not Met: The action is valid but cannot be gain
progress because a necessary prior condition is not met (e.g.,
trying to 'put' an object that is not currently held).
```

---

```
- Condition Met, Action Not Taken: A critical step was available,
necessary to progress and the agent already observed the condition,
but the agent performed an irrelevant or less optimal action
instead (e.g., finding the target object but not picking it up).
- Incorrect Target: The action is performed on the wrong object or
at the wrong location (e.g., picking up a 'cloth' instead of the
required 'soapbottle').
- Repeated Action: The action is part of a sequence that was
already performed and without any progress.\n """

for shot in shots:
    prompt += f"Example:\nInput: {shot['input']}\n
            Label: {shot['label']}\n\n"
prompt += f"Input: {input_text}\nLabel:"
```

**RAG prompt for Error Detection**

```
prompt = "You are a classifier. Given the following examples,
predict the label for the new sample.\n"
for idx, (txt, lbl) in enumerate(retrieved):
    prompt += f"Example {idx+1}:\nText: {txt}\nLabel: {lbl}\n"
prompt += f"\nNow,classify this sample:\nText: {X_test[i]}\nLabel: "
```

## D  LLM USAGE

In this paper, we leverage LLMs, including ChatGPT and Gemini, to refine sentence-level writing.