# OpenReview forum: "Trajectory Graph Copilot: Pre-Action Error Diagnosis in LLM Agents"
_ICLR.cc/2026/Conference — Submitted to ICLR 2026_

### Official Review · Reviewer_B51S · 2025-10-29

**Soundness:** 2
**Presentation:** 3
**Contribution:** 2
**Rating:** 4
**Confidence:** 4

**Summary:**

This paper aims to address the issue of compounding errors in LLM agents, where a single suboptimal action can derail an entire trajectory. To achieve step-level action error detection, the authors draw inspiration from software debugging and propose a graph-based method called Trajectory Graph Copilot. They use actions as nodes and observations as edges to construct a graph, then train a mapping function to probe potential errors through GNNs. Experiments conducted across representative LLM agent environments (AlfWorld, TextWorld, ScienceWorld, and TravelPlanner) validate the effectiveness of the proposed method in error detection.

Overall, I think this paper targets an important problem in LLM agents. However, the contributions to methodology in this work are still insufficient.

**Strengths:**

1. This paper targets error diagnosis in LLM agents and formulates a probabilistic graph model to address the step-level action error detection. The key idea is interesting.

2. The proposed Trajectory Graph Copilot pipeline makes sense.

3. The overall logical flow for paper writing is clear.

**Weaknesses:**

1. This work constructs an action-centric graph, employs GNNs with message passing, and regards action error detection as a trajectory node classification task. The core contributions appear limited when compared to several related works that also employ graph-based approaches [1–2] or GNNs [3].

2. The relationship between Gebugger and Trajectory Graph Copilot is unclear. While Gebugger is described as a diagnostic module for error detection in the Introduction, it is referred to as the graph construction component in Section 3, which creates confusion.

3. The authors claim to "shift the paradigm from post-hoc trajectory analysis to proactive, step-level error diagnosis" (Lines 49–50). However, describing the process as "debugging" may be misleading, as it still relies on post-hoc analysis of executed trajectories. A more appropriate analogy might be a "compiler," which performs pre-action error checking, aligning better with the intended proactive approach.

4. It is difficult to see how the theoretical analysis in Section 4 directly addresses any specific concerns related to the proposed method. Moreover, this analysis is not supported by experimental validation, which weakens its impact and makes the section less convincing and insightful.


[1] Lee et al., DHRL: A Graph-Based Approach for Long-Horizon and Sparse Hierarchical Reinforcement Learning. NeurIPS'2022.
[2] Yu et al., Scene Graph-Guided Proactive Replanning for Failure-Resilient Embodied Agents. RSS'2024.
[3] Wu et al., Can Graph Learning Improve Planning in LLM-based Agents? NeurIPS'2024.

**Questions:**

1. The objective of the paper is somewhat unclear. Is the focus solely on error detection, or does it also include error correction? Additionally, the term "feedback evaluation" lacks a clear definition: what exactly is being evaluated and how?

2. The experimental setup involves using LLMs to generate step-level error detection labels, followed by manual selection and filtering. If LLMs are already capable of generating these labels, what is the added value or necessity of introducing the Trajectory Graph Copilot for error detection?

3. Potential issues may occur where an action or observation is outside the defined scope. How does "we use embeddings to search for and retrieve the corresponding trajectory path from the existing graph" (Lines 216-217) work?

4. Experimental results in Table 1 are somewhat confusing. Each dataset is associated with three columns representing different backbone LLMs. It is unclear how these columns relate to the LLMs mentioned in the rows. What's the difference between column-wise LLMs and row-wise backbones?

---

> ### Author Response · Authors · 2025-11-20
> **Response to reviewer B51S(part 1/3)**
>
> Dear Reviewer B51S,
>
> Thank you for your careful reading and insightful suggestions. We answer the questions below:
>
> > W1. This work constructs an action-centric graph, employs GNNs with message passing, and regards action error detection as a trajectory node classification task. The core contributions appear limited when compared to several related works that also employ graph-based approaches [1–2] or GNNs [3].
>
> Thanks for the valuable comments. In our paper, the core contributions consist of two parts: (1) introducing graph-based techniques for action error detection from the perspective of a partially observable Markov decision process (POMDP), and (2) using action error detection as a sandbox to support LLM-Agent decision making.
>
> The references mentioned by the reviewer mainly focus on reinforcement learning and planning, which fall outside the scope of our work. Our paper does not involve reinforcement learning algorithms or agent-planning methodologies. Therefore, we believe direct comparison with those references would not be appropriate or fair, as they target fundamentally different problems. Thanks for introducing the valuable works; we have added them to the related works.
>
>
>
> > W2. The relationship between Gebugger and Trajectory Graph Copilot is unclear. While Gebugger is described as a diagnostic module for error detection in the Introduction, it is referred to as the graph construction component in Section 3, which creates confusion.
>
>
> Thanks for the insightful suggestions. In our work, Gebugger refers specifically to the action error detection module, which is composed of the graph construction component and the GNN model. The Trajectory Graph Copilot represents the full pipeline, where Gebugger is used to generate feedback that guides LLM-Agent decision-making. In Figure 2, the entire pipeline illustrates the framework of the Trajectory Graph Copilot, while the component shown in the upper-right corner corresponds to Gebugger. We have revised the manuscript to clarify this distinction and make the framework easier to understand.
>
>
> > W3. The authors claim to "shift the paradigm from post-hoc trajectory analysis to proactive, step-level error diagnosis" (Lines 49–50). However, describing the process as "debugging" may be misleading, as it still relies on post-hoc analysis of executed trajectories. A more appropriate analogy might be a "compiler," which performs pre-action error checking, aligning better with the intended proactive approach.
>
> Thanks for pointing out this question. While “Compiler” is a reasonable alternative, we believe “Debugging” is a more suitable description for our setting. In this paper, we use the name Gebugger for two main reasons. First, the module operates on actions and their labels. Similar to how a human expert inspects each step in a procedure, our module checks every action in the trajectory. The name “Gebugger” reflects this step-by-step inspection process based on graph representations.  Second, at test time, the goal of proactive diagnosis is to improve the performance of LLM agents. Although it is possible to run the checking process only after the entire trajectory has been generated, its effectiveness would be much weaker because the decisions have already been made. This is analogous to how debuggers help identify logical issues during program execution rather than evaluating whether the final program is well-written without grammatical issues. Based on these considerations, we believe “Gebugger” is an appropriate and meaningful name for our module.

---

> ### Author Response · Authors · 2025-11-20
> **Response to reviewer B51S(part 2/3)**
>
> > W4. It is difficult to see how the theoretical analysis in Section 4 directly addresses any specific concerns related to the proposed method. Moreover, this analysis is not supported by experimental validation, which weakens its impact and makes the section less convincing and insightful.
>
> Thank you for the thoughtful comment. In Section 4, we aim to provide theoretical insight showing that graph-based detection methods have advantages over sequence-based approaches. Our analysis is based on the POMDP (Partially Observable Markov Decision Process) formulation of agent trajectories.
>
> We begin with the Markov Blanket assumption, which states that the label of each action depends only on a limited set of previous steps rather than the entire trajectory. This is consistent with how action correctness is determined in interactive environments.
>
> For example, consider the task “pick an apple and place it on the table.” Once the agent performs the action “pick up apple,” the correctness of subsequent actions depends primarily on whether the apple has been picked. If the next step is “go to refrigerator,” then:
>
> - if the agent already holds an apple, this step is an incorrect target (the agent should place the apple on the table);
>
> - if the agent does not hold an apple, the same step is correct, because the agent may be searching for an apple.
>
> In both cases, only a small and localized set of previous actions is required to determine the label. Therefore, assuming a Markov Blanket over a limited neighborhood is reasonable for agent decision-making tasks.
>
> Next, we argue that sequence encoders may introduce environment-related noise. When encoding the entire trajectory as a long sequence, irrelevant early actions are processed together with the relevant context， such as those occurring before “pick up apple”. This global encoding can dilute or distort the information truly needed for classification, making the detection task harder.
>
> Based on these assumptions, we show that graph-based methods can provide a tighter lower bound for the detection task. By explicitly modeling only the relevant dependencies and removing irrelevant temporal noise, graph structures offer a more efficient representation of action–state relations. As a result, for the same detection error, graph-based methods require fewer samples than sequence-based approaches.
>
> Finally, our empirical action-error detection results support this theoretical motivation: graph-based models consistently outperform sequence-based baselines under the same settings.
>
>
> > Q1. The objective of the paper is somewhat unclear. Is the focus solely on error detection, or does it also include error correction? Additionally, the term "feedback evaluation" lacks a clear definition: what exactly is being evaluated and how?
>
>
> Thank you for raising these constructive questions. Error correction is indeed an interesting direction, but it is beyond the scope of this paper. Effective error correction requires strong reasoning capabilities, while our graph model is not designed to perform such reasoning. Therefore, in this work, our focus is limited to detecting errors and providing feedback signals, rather than correcting the actions themselves.
>
> In this paper, the “feedback evaluation” specifically examines whether the detection-based feedback can improve LLM-Agent decision-making. The evaluation metric is the success ratio, where we compare the performance of a vanilla agent against the version enhanced with our feedback. As the results show, providing feedback based on our detection module consistently improves the agent’s performance.
>
> Thank you again for the question. To avoid ambiguity, we have revised the corresponding parts in the manuscript for clarity.
>
>
> > Q2. The experimental setup involves using LLMs to generate step-level error detection labels, followed by manual selection and filtering. If LLMs are already capable of generating these labels, what is the added value or necessity of introducing the Trajectory Graph Copilot for error detection?
>
> We thank the reviewer for this insightful question. During the dataset construction phase, we first collect trajectories for the training tasks. Since each training task includes a corresponding expert/golden path, we can use these references to accurately annotate labels via an LLM. However, for the test set, such golden paths are unavailable, meaning an LLM would have to rely solely on its internal reasoning to detect errors. This limitation necessitates the introduction of our Trajectory Graph Copilot, which provides the required structure for accurate inference without ground truth.

---

> ### Author Response · Authors · 2025-11-20
> **Response to reviewer B51S(part 3/3)**
>
> > Q3. Potential issues may occur where an action or observation is outside the defined scope. How does "we use embeddings to search for and retrieve the corresponding trajectory path from the existing graph" (Lines 216-217) work?
>
> Thanks for the valuable question. To clarify, our framework employs an action-centric graph, where nodes represent actions and edges correspond to observations. Ideally, the training set would cover the entire action space; however, to handle unseen or out-of-scope actions during inference, we utilize a similarity-based approach. Specifically, we map the current action to the most semantically similar node in the graph, effectively treating the problem as a standard node classification task.  There are some other options, such as using an "unknow node" for out of scope during training, ignoring the action.
>
>
> > Q4. Experimental results in Table 1 are somewhat confusing. Each dataset is associated with three columns representing different backbone LLMs. It is unclear how these columns relate to the LLMs mentioned in the rows. What's the difference between column-wise LLMs and row-wise backbones?
>
> We thank the reviewer for this valuable question.  In Table 1, for each environment, we evaluate three LLM-Agents on the tasks. Each column corresponds to one LLM-Agent within a specific environment. The rows correspond to different methods, which are grouped into four categories. Within each category, we use different LLM backbones. For example, in the RAG methods, we use three different LLMs for answer generation: GPT-4o, Qwen2.5, and Gemma3. Importantly, there is no performance difference caused by the choice of backbone within the same method. We appreciate the comment and will revise this part in the manuscript to make the table and its explanation clearer.
>
> We sincerely thank the reviewer for your time and thoughtful feedback. Your comments have helped us identify important areas for clarification and improvement in our paper.

---

> ### Comment · Reviewer_B51S · 2025-11-26
>
> Thank you for your detailed responses. After careful consideration, I remain unconvinced that the graph-based techniques for action error detection constitute a major technical contribution. My earlier concerns about the paper’s objectives and experimental design have been largely resolved. In light of these clarifications, I have increased my overall score accordingly.

---

> ### Author Response · Authors · 2025-11-26
> **Reply to reviewer B51S**
>
> Dear Reviewer B51S,
>
> Thank you for re-engaging with our paper and raising your overall score. We appreciate your acknowledgment that our clarifications helped resolve the concerns regarding the paper’s objectives and experimental design. Regarding the remaining concern about the contribution, while we agree that the individual components are based on established techniques, our contribution lies in the formulation and integration of these components, rather than in proposing a new graph algorithm. Specifically:
>
> - We formulate step-level proactive error diagnosis as a graph node error detection problem, where the error label space is grounded in the POMDP.  We also provide theoretical and empirical analysis showing the advantages of the graph-based approach over standard retrieval-based methods (e.g., retrieval, RAG).
>
> -  We develop a framework that integrates step-level proactive error diagnosis as a sandbox tool for agent decision-making. This reframing, which converts post-hoc trajectory reward to pre-action error detection, is new in this domain.
>
> - We validate the effectiveness of this formulation through extensive experiments. The 14.69% average improvement across four diverse benchmarks and three LLM agents shows that the proposed integration is practically beneficial while requiring no model fine-tuning.
>
> We hope this framing clarifies the intention behind our contribution. We sincerely appreciate your thoughtful feedback and the time you dedicated to reviewing our work.
>
> Best,
> Authors

---

### Official Review · Reviewer_V4Co · 2025-10-31

**Soundness:** 3
**Presentation:** 3
**Contribution:** 3
**Rating:** 6
**Confidence:** 4

**Summary:**

The paper “Trajectory Graph Copilot” introduces GEBUGGER, a probabilistic graph-based diagnostic module designed to detect step-level action errors in LLM-based agents performing long-horizon tasks. The key insight is that an LLM agent’s interaction sequence (observation–action–reward trajectory) can be represented as a graph, where nodes are actions and edges encode contextual transitions via observation embeddings.

**Strengths:**

- The motivation presented in Lines 44–46 is clear and convincing. The authors correctly identify that learning solely from full-trajectory outcomes provides weak causal signals and fails to pinpoint the origins of errors within long-horizon tasks.

- Its analogy to software debugging is conceptually strong and intuitive. Shifting the paradigm from post-hoc trajectory evaluation to proactive action analysis is an interesting idea.

- The paper provides a solid theoretical justification showing that graph-based representations achieve lower Bayes risk and improved sample efficiency compared to sequence-based models, which lends analytical depth to the framework.

- The graph-based approach not only achieves higher performance with fewer samples but also functions as a diagnostic sandbox for LLM agents, allowing them to self-correct based on structured error signals. The proposed GEBUGGER module is comprehensively evaluated across multiple environments and agents, consistently demonstrating superior performance.

- Empirically, the framework exhibits robust and significant gains, achieving an average 14.69% improvement in pass ratio across diverse embodied and text-based environments (AlfWorld, ScienceWorld, TextWorld, and TravelPlanner).

**Weaknesses:**

- The experiments seem to be conducted only under in-distribution setups (e.g., training and testing on the same environment such as AlfWorld or ScienceWorld). It remains unclear how well the proposed framework generalizes to out-of-distribution settings, which is critical since classification-based agents often overfit to specific training domains.

- More ablation and sensitivity analyses are needed to isolate the contribution of each module in the proposed pipeline. For instance, how does the model behave when the in-context feedback is intentionally perturbed or incorrect? Such studies would clarify the robustness and causal contribution of each component.

- The paper introduces six error categories (Appendix B.1) but does not provide sufficient intuition or empirical justification for their design. How were these categories derived, and how exactly does the model distinguish among them?

- The term “step-level” is used repeatedly (e.g., step-level diagnosis, step-level feedback), yet a formal definition is missing. Clarifying whether it strictly refers to each action-observation pair or another temporal granularity would improve precision.

- Notation clarity can be improved. For example, the variable a is used for both action pairs and aggregated messages, and the meaning of T in Equation (1) is ambiguous (possibly referring to trajectory sets). Additionally, the interpretation of v_j \in T but not \in A needs clarification.

- The paper mentions “more generalized representation” and “partial observation” (L150–151) but does not concretely describe how these notions are implemented or measured within the model.

- The baseline selection could be modernized. Comparing primarily against TF-IDF (1988) and BERT (2019) limits the empirical impact; including or at least discussing more recent GNN-based or retrieval-augmented approaches would strengthen the experimental section.

**Questions:**

Please see the weaknesses.

---

> ### Author Response · Authors · 2025-11-20
> **Response to reviewer V4Co(part 1/2)**
>
> Dear Reviewer V4Co,
>
> We sincerely appreciate your feedback and comments on our paper.  We provide the responses below:
>
> > W1. The experiments seem to be conducted only under in-distribution setups (e.g., training and testing on the same environment such as AlfWorld or ScienceWorld). It remains unclear how well the proposed framework generalizes to out-of-distribution settings, which is critical since classification-based agents often overfit to specific training domains.
>
> Thank you for the insightful comments. In our experiments, the learned knowledge is domain-specific. For instance, knowledge acquired from training tasks in AlfWorld may not be directly applicable to TravelPlanner. In our setup, we explicitly consider unseen tasks. In AlfWorld, the test set contains tasks and rooms that do not appear in the training set. In ScienceWorld, while the tasks may overlap between training and test sets, the environments themselves are unseen.
> Therefore, we believe that under the same task distribution, our method generalizes well to both unseen environments and unseen tasks. To clarify this point, we will revise the corresponding section in the manuscript.
>
>
> > W2. More ablation and sensitivity analyses are needed to isolate the contribution of each module in the proposed pipeline. For instance, how does the model behave when the in-context feedback is intentionally perturbed or incorrect? Such studies would clarify the robustness and causal contribution of each component.
>
> Thank you for the valuable suggestions and comments. In our experiments, we also consider replacing our detector with alternative methods, such as TF-IDF. For graph construction, we investigate different graph categories, including variations in node features and edge types.
>
> To address concerns regarding sensitivity, we include an ablation study on hyperparameters in Section 6.3 and Appendix C.3, such as maximum attempts and confidence thresholds. To further verify the robustness of our method, we conduct additional experiments in the TextWorld environment. Specifically, we randomly inject noise into the error detection results and observe the resulting changes in performance. The results are provided in the table below. As shown, performance decreases as noise increases, demonstrating the effectiveness and reliability of our detection module.
>
> We sincerely thank the reviewers for the insightful comments and have revised the corresponding parts of the manuscript accordingly.
>
>
> | Noise | Qwen2.5(Pass Ratio%) |  Gemma3(Pass Ratio %)|
> |-------|----------------------|----------------------|
> |0.0| 65.00 | 75.00|
> |0.2| 57.00 | 75.00|
> |0.4| 60.00 | 75.00|
> |0.6| 55.00 | 67.50|
> |0.8| 55.00 | 70.00|
> |1.0| 55.00 | 67.50|
>
>
> > W3. The paper introduces six error categories (Appendix B.1) but does not provide sufficient intuition or empirical justification for their design. How were these categories derived, and how exactly does the model distinguish among them?
>
>
> Thank you for the insights regarding the error categories. These categories are defined based on the experience and domain knowledge of the agent. In AgentBoard, tasks are decomposed into subgoals. Inspired by this idea, we borrow the notion of state from MDP (Markov Decision Process) to define milestone states. Based on these milestone states, we then define general error categories that are independent of any specific environment or available tools.
>
> In our dataset, each task has a corresponding environment with a golden path, which we use as a reference to annotate each action. During training, the model learns patterns from these labels across the graph. We clarified this process in the dataset construction section of the manuscript.
>
>
> > W4. The term “step-level” is used repeatedly (e.g., step-level diagnosis, step-level feedback), yet a formal definition is missing. Clarifying whether it strictly refers to each action-observation pair or another temporal granularity would improve precision.
>
> Thank you for pointing out this issue. In our work, one step corresponds to one action, and therefore “step-level” is equivalent to “action-level.” Our method provides error detection for the current action, rather than evaluating the correctness of the entire trajectory. We will revise the manuscript to clarify this point.

---

> ### Author Response · Authors · 2025-11-20
> **Response to reviewer V4Co(part 2/2)**
>
> > W5. Notation clarity can be improved. For example, the variable a is used for both action pairs and aggregated messages, and the meaning of T in Equation (1) is ambiguous (possibly referring to trajectory sets). Additionally, the interpretation of v_j \in T but not \in A needs clarification.
>
> Thanks for providing the writting issues. In Eq. 1, the symbol $t$ originally refer to one trajectory and $\mathcal{T}$ denotes the set of trajectories. However, since $t$ is used to represent a timestamp, we have revised this part to use a new letter $\xi$ to represent a trajectory to avoid confusion.
> There are two types of nodes in our graph: action nodes and trajectory nodes, as illustrated in Figure 2. Specifically, $v \in \mathcal{A}$ indicates $v$ is an action node, otherwise $v \in \mathcal{T}$ indicates that $v$ represent a trajectory. Thanks again for the comments, we fixed the typo and revised paper to make it easy to understand.
>
>
> > W6. The paper mentions “more generalized representation” and “partial observation” (L150–151) but does not concretely describe how these notions are implemented or measured within the model.
>
> We thank reviwer for the insights about the generalization issues. Lines 148–151 discuss the use of a heterogeneous graph to represent POMDP (Partially Observable Markov Decision Process) state transitions.  This method use two kinds of nodes, action and states, making it more effective in a limited action space environment. Our method is inspired by this idea; however, we do not directly adopt it due to the difficulty of accurately estimating the environment states. We will clarify this part to make it easy to read and understand.
>
>
> > W7. The baseline selection could be modernized. Comparing primarily against TF-IDF (1988) and BERT (2019) limits the empirical impact; including or at least discussing more recent GNN-based or retrieval-augmented approaches would strengthen the experimental section.
>
> Thank you for the baseline suggestions. In our experiments, we use RAG (retrieval-augmented generation) as a baseline. As shown in Table 1 and Figure 3, our method outperforms the RAG approaches.
>
> Regarding other GNN-based methods, we acknowledge that modern GNN techniques may achieve better performance. However, we would like to emphasize that Trajectory Graph Copilot is a unified framework in which the GNN backbone can be replaced. In our experiments, we only use the classical GCN as the backbone.
>
> To further address the concern about GNN-based methods, we include  an additional comparison with TextGCN[1], a document classification techique, on TextWorld. For TextGCN, we use a two-layer GCN as the architecture with Adam optimizer, a learning rate of 1E-3, weight decay 1E-4, and 2000 epoches. We report the detection accuracy(%) as below. As the results show, TextGCN does not perform well. We attribute this to the fact that TextGCN is designed for capture words co-occurance patterns instead of the action dependency, which are crucial for our task.
>
> | Method | GPT4o-mini | Qwen2.5 | Gemma3|
> | -----  | ---------- | ------- | ----- |
> | TextGCN | 59.77 $\pm$ 0.17 | 53.69 $\pm$ 0.80 | 54.24 $\pm$ 4.41|
> | Ours   | 68.72 $\pm$ 0.91 | 58.10 $\pm$ 1.33 | 63.16 $\pm$ 2.29 |
>
> reference:
>
> [1]. Graph Convolutional Networks for Text Classification, AAAI 2019
>
> We thank you again for your constructive comments and for your efforts to improve the quality of our paper.

---

### Official Review · Reviewer_6EJw · 2025-11-12

**Soundness:** 2
**Presentation:** 3
**Contribution:** 2
**Rating:** 4
**Confidence:** 3

**Summary:**

LLM-based agents perform impressively but remain brittle on long-horizon interactive tasks such as embodied AI or complex planning. Small early mistakes can cascade into full-trajectory failures. Existing methods mostly apply post-hoc refinement or trajectory-level rewards, offering little insight into which step-level actions caused failures. Trajectory Graph Copilot is introduced to proactively diagnoses potential action errors before execution. Its core module, GEBUGGER, represents past trajectories as a probabilistic graph and employs a GNN to detect patterns associated with failure. It then issues early-warning signals so the LLM can self-correct via reasoning. The paper then provides a theoretical analysis on the bayes risks under two assumptions for graph-based representations and sequence-based representations. Experiments on multiple tasks reveals that GEBUGGER consistently outperforms text-classification, retrieval and RAG-based baselines.

**Strengths:**

- The paper provides a novel conceptual framework that reframes error diagnosis in LLM agents from post-hoc trajectory repair to pre-execution error prediction. The proactive copilot design provides real-time diagnostic warnings before the agent acts, preventing cascaded errors in long-horizon tasks.
- The paper provides comprehensive experimental results on four benchmarks and show consistent gains.
- The paper provides detailed ablation studies confirming design choices and stability where the results show directed BERT-based graphs perform best and that feedback integration steadily increases task success.

**Weaknesses:**

- **Synthesized labels from stronger models and manual labeling:** The major concern of this paper is that it involves a pre-defined label space with LLM synthesized labels as supervision. The paper even reported manual filtering on the generated labels which reduces the transparency of the approach (see line 896-899). Apart from these, the approach relies on other complicated engineering. For example, they reported utilizing preprocessing on the texts (see line 158-160). All these could undermine the effectiveness of the proposed method and requires further experimental comparisons to solidify.
- **Assumption 2 too strong without evidence:** The theoretical analysis is depended on assumption 2 where it states that “the sequence representation U is not a deterministic function of S” and “… correlated with environment-specific artifacts and not conditionally independent of Y given X”. This indeed is too strong and might depend on model long-context capability. In contrast to BERT models used as a baseline in the paper, models like Qwen or Llama might be a more robust “context filter” that extracts relevant information from the environment feedbacks. It requires experimental validation to confirm.
- **Unequal amount of supervisions used for baselines and GEBUGGER:** The comparison in experiments reveals potential fairness and supervision imbalance. GEBUGGER uses a graph-constructed dataset with explicit step-level error labels synthesized and curated, while several baselines (e.g. RAG, retrieval, LLM-as-judge) rely on few-shot or zero-shot inference without labeled training. Thus, the claimed accuracy gap might not come from architectural superiority.

**Questions:**

N/A

---

> ### Author Response · Authors · 2025-11-20
> **Response to reviewer 6EJw(part 1/2)**
>
> Dear Reviwer 6EJw,
>
> Thank you very much for your valuable feedback. We appreciate your time and effort. Our responses are provided below.
>
> > W1. Synthesized labels from stronger models and manual labeling: The major concern of this paper is that it involves a pre-defined label space with LLM synthesized labels as supervision. The paper even reported manual filtering on the generated labels which reduces the transparency of the approach (see line 896-899). Apart from these, the approach relies on other complicated engineering. For example, they reported utilizing preprocessing on the texts (see line 158-160). All these could undermine the effectiveness of the proposed method and requires further experimental comparisons to solidify.
>
> Thanks for bringing these concerns. To improve the transparency of our approach, we will release the raw data, processed data, and code. In Section 2 and Appendix B.1, we define the error space based on expert experience, inspired by the AgentBoard benchmark, which decomposes each task into several subgoals. Using these definitions, we first employ an LLM to generate initial labels. We then manually review and correct the labels to ensure dataset quality. The golden paths of the training tasks serve as the reference for our label annotation process. We have revised the manuscript to clarify this procedure.
>
> Regarding our method, we rely on standard tools for routine preprocessing; for example, using NLTK for text preprocessing is common in NLP workflows. Importantly, this part is not central to action error detection — the key component is the graph construction process. For graph construction, we consider two edge (or node feature) types and conduct an ablation study in Section 6.3. To further address the reviewer’s concern, we have carefully checked and revised the relevant descriptions in the paper.
>
>
> > W2. Assumption 2 too strong without evidence: The theoretical analysis is depended on assumption 2 where it states that “the sequence representation U is not a deterministic function of S” and “… correlated with environment-specific artifacts and not conditionally independent of Y given X”. This indeed is too strong and might depend on model long-context capability. In contrast to BERT models used as a baseline in the paper, models like Qwen or Llama might be a more robust “context filter” that extracts relevant information from the environment feedbacks. It requires experimental validation to confirm.
>
> Thanks for pointing out this concern. To make it more accuracte, we have revised the assumption 2. In this statement, the embedding $U$ is extract by a well-pretrained model $\phi$ with the input sequence $X$. In general, the embedding $U$ is influenced by all the tokens/words in $X$. However, it would introduce the environment noise.
>
> To address the concern of model capability, we conduct the experiments on the backbone of text classification method on TextWorld. We use Qwen2.5-0.5B as the backbone, with AdamW as optimizer, learning rate 2E-5 for 10 epoches. We report the accuracy(%) of five experiment results as follows. As the result show, with a larger backbone, the detection perforamnce increases. For GPT4o-mini, Qwen2.5-14B as the agent, it have a similar performance to ours. While as the Gemma3-27B, our method still have advantage than sequence-based methods. The results are algined with our theoritical analysis.
>
> | Method |  GPT4o. | Qwen2.5 | Gemma3|
> |--------|---------|---------|-------|
> | Bert| 63.68 $\pm$ 0.97 | 49.96 $\pm$5.01 | 43.45 $\pm$ 4.93 |
> | Qwen2.5-0.5B| **69.19 $\pm$ 1.45**| 57.78 $\pm$ 3.72 | 56.89 $\pm$ 4.25|
> | Ours| 68.72 $\pm$ 0.91 | **58.10 $\pm$ 1.33** | **63.16 $\pm$ 2.29**|

---

> ### Author Response · Authors · 2025-11-20
> **Response to reviewer 6EJw(part 2/2)**
>
> > W3. Unequal amount of supervisions used for baselines and GEBUGGER: The comparison in experiments reveals potential fairness and supervision imbalance. GEBUGGER uses a graph-constructed dataset with explicit step-level error labels synthesized and curated, while several baselines (e.g. RAG, retrieval, LLM-as-judge) rely on few-shot or zero-shot inference without labeled training. Thus, the claimed accuracy gap might not come from architectural superiority.
>
> Thanks for the insightful comments. In our experiments, all training-based methods follow the same dataset split. Although the labels in the dataset are imbalanced, the label distribution is consistent across the training, validation, and test sets. Therefore, we believe the comparison among training-based methods is fair.
>
> For the training-free baselines, including retrieval, RAG, and LLM-as-judge, the labels come from the database or the prompt. In the retrieval and RAG baselines, we use the entire training set as the labeled database. Given a test instance, these methods retrieve the most similar trajectories along with their labels. For the LLM-as-judge baselines, we provide the definition of the label space, and in the one-shot/three-shot settings, we also include example instances for each label. Thus, we consider the comparison between our method and these baselines to be fair as well. To further clarify the experimental setup, we have added detailed descriptions in the Appendix.
>
>
> We sincerely thank the reviewer for your time and thoughtful feedback. Your comments have helped us identify important areas for clarification and improvement in our paper.

---

> ### Author Response · Authors · 2025-11-27
> **Looking forward to your reply**
>
> Dear Reviewer 6EJw,
>
> We greatly appreciate the time you took to review our paper.  As the review deadline approaches, we would greatly appreciate any feedback you could provide on the revised submission at your earliest convenience. Your insights are invaluable to us, and we sincerely welcome any additional comments or suggestions you may have. We are always willing to provide additional information or clarification if necessary.
>
> Best,
>
> The Authors

---

### Official Review · Reviewer_ZAio · 2025-11-12

**Soundness:** 2
**Presentation:** 3
**Contribution:** 2
**Rating:** 6
**Confidence:** 4

**Summary:**

This paper introduces Trajectory Graph Copilot (TGC), a framework that proactively diagnoses potential errors in LLM agent actions before execution. The core component, GEBUGGER, models historical trajectories as probabilistic graphs using Graph Neural Networks (GNNs) to identify action patterns associated with failure. Instead of post-hoc corrections or fine-tuning, TGC acts as a “diagnostic sandbox,” warning agents about risky actions and prompting self-correction. Experiments on four benchmarks (AlfWorld, TextWorld, ScienceWorld, and TravelPlanner) and three LLMs (GPT-4o-mini, Qwen2.5, Gemma3) show a 14.69% average improvement in task completion rates over baselines.

**Strengths:**

1. This method shifts from post-hoc to pre-action error diagnosis, which is a useful conceptual advance.

2. Provides theoretical analysis demonstrating lower Bayes risk and sample complexity compared to sequence-based baselines.

3. Tests across multiple environments and agents, showing consistent improvements.

4. The “copilot” design fits naturally into existing LLM-agent frameworks without retraining the base model.

**Weaknesses:**

1. There is lack of explicit explanations of predicted errors, and why this method systematicaly improves overthe baseline. It would be good to show some quantative examples how this method makes improvement.

2. This method requires labeled trajectory datasets with fine-grained error annotations, which may be difficult to scale.

3. Some comparisons lack detailed justification or variance analysis, what's the variance of the results?

4. While graph structure variants are tested, the effects of feedback integration mechanisms (e.g., prompt formatting, feedback text) are only briefly analyzed.

**Questions:**

Will more delibrate prompting-based LLM agent baselines perform as well as adding this GNN-based prediction module?

---

> ### Author Response · Authors · 2025-11-20
> **Response to reviewer ZAio(part 1/2)**
>
> Dear Reviewer ZAio,
>
> Thank you very much for taking the time to review our manuscript and providing insightful and valuable comments. Your feedback has been incredibly helpful in improving the quality of this work.
>
>
> > W1. There is lack of explicit explanations of predicted errors, and why this method systematicaly improves overthe baseline. It would be good to show some quantative examples how this method makes improvement.
>
> Thanks for the insightful comments. In this paper, inspired by the POMDP(Partially Observation Markov Decision Process), we use the graph to represent the trajectories. In section 4, we discusse the advantage of graph based methods than sequence based methods. We make the theoritical analysis based on two assumptions. First,  the Markov Blanket, the action label only related to a limited steps. Second, the sequence based model would capture the environment noise. Then we find that the graph based method could reach a lower error bound and require less samples given a error ratio. We provide an example to explain why the markov blanket is reasonable.
>
> For example, given a task "pick an apple and place on table", once the agent picked an apple, the following action labels only need to backtrace to the action "pick up apple [num]". If the agent the next step is "go to refrigerator [num]", with apple already picked up, this action's label should be incorrect target. However, if the agent do not have an apple, then this action should be no error, because it is looking for an apple.  In this case, we only need a limited steps to make the prediction. So Markov Blanket assumption is reasonable in the agent decsion making.
>
> Thanks again to bright this weakness. we revised our paper to make it easy to understand.
>
>
>
> > W2. This method requires labeled trajectory datasets with fine-grained error annotations, which may be difficult to scale.
>
> Thanks for brining the concern about scale. In our experiments, it requires golden paths or milestone states to estimate the action labels. With the advantage of graph, we show that graph based methods require less samples than the sequence based methods. We still have advantage than traditional retrieval based methods. We admit that the label annotations is a limitation of this method. We leave it for the future work. Thanks again for pointing out the insightful concerns.
>
>
> > W3. Some comparisons lack detailed justification or variance analysis, what's the variance of the results?
>
> Thank you for the valuable comments. To address this issue, we added experiments reporting the error bars for both the action-error detection and feedback-improvement evaluations on Textworld environment. We repeat each experiment five times and report the mean and standard deviation. The results are provided in the two tables below. In these experiments, the retrieval model is GTR-T5-Large (GTR), and the LLM-based methods use Gemma3-27B. As shown by the results, the performance trends are consistent with those reported in the paper. We have revised the manuscript accordingly. Thank you again for the insightful feedback.
>
> | Method |  Acc/Micro F1 (Qwen2.5) | Acc/Micro F1 (Gemma3) |Macro F1 (Qwen2.5) | Macro F1 (Gemma3) |
> | :--- | :---: | :---: | :---: | :---: |
> | TF-IDF |  *0.4996 ± 0.0041* | 0.4045 ± 0.0160 |  0.2712 ± 0.0042 | 0.2256 ± 0.0074 |
> | Bert |  *0.4996 ± 0.0501* | 0.4345 ± 0.0493 |  0.3897 ± 0.0380 | 0.3711 ± 0.0283 |
> | Retrieve| 0.3018 ± 0.0009 | 0.5322 ± 0.0073 |  0.2472 ± 0.0056 | 0.4257 ± 0.0041 |
> | RAG |  0.5183 ± 0.0066 | *0.6124 ± 0.0072* |  **0.4649 ± 0.0036** | 0.4952 ± 0.0052 |
> | LLM Zero-Shot  |  0.3427 ± 0.0029 | 0.4492 ± 0.0000 |  0.2462 ± 0.0009 | 0.2680 ± 0.0000 |
> | LLM One-Shot |  0.2620 ± 0.0741 | 0.3090 ± 0.0554 |  0.1824 ± 0.0606 | 0.1799 ± 0.0442 |
> | LLM Three-Shot |  0.3194 ± 0.0505 | 0.1876 ± 0.0907 |  0.1932 ± 0.0612 | 0.1474 ± 0.0694 |
> | **Ours** | **0.5810 ± 0.0133** | **0.6316 ± 0.0229** |  *0.4298 ± 0.0063* | **0.5283 ± 0.0199** |
>
>
> | Method | Pass Ratio(%) (Qwen2.5) | Pass Ratio(%) (Gemma3) | Ground Ratio(%) (Qwen2.5) | Ground Ratio(%) (Gemma3) |
> | :--- | :---: | :---: | :---: | :---: |
> | Vanilla | 65.00 ± 6.52 | 70.50 ± 5.50 | **88.77 ± 2.98** | **91.49 ± 3.42** |
> | +TF-IDF | 61.50 ± 3.35 | 67.50 ± 7.07 | 40.49 ± 2.35 | 46.81 ± 1.51 |
> | +Retrieve | 67.50 ± 6.12 | 73.50 ± 6.75 | 41.16 ± 1.73 | 51.52 ± 1.13 |
> | +RAG | **69.50 ± 3.71** | 71.94 ± 5.36 | 40.81 ± 0.82 | 52.66 ± 1.42 |
> | **+Ours** | 64.00 ± 1.22 | **76.00 ± 3.74** | 73.61 ± 6.74 | 87.15 ± 4.78 |

---

> ### Author Response · Authors · 2025-11-20
> **Response to reviewer ZAio(part 2/2)**
>
> > W4. While graph structure variants are tested, the effects of feedback integration mechanisms (e.g., prompt formatting, feedback text) are only briefly analyzed.
>
> Thank you for the insights regarding the integration mechanisms. To address this concern, we added an ablation study on this aspect. We evaluate four types of integration mechanisms: (1) incorporating feedback in the user prompt, (2) including feedback in the user prompt without label explanations, (3) adding feedback in the system prompt, and (4) appending feedback to the trajectory history. We conduct these experiments in the TextWorld environment, and the results are provided in the table below.
>
> As shown in the results, Gemma3-27B is more robust than Qwen2.5-14B across different feedback integration strategies. Another observation is that a high grounding ratio does not necessarily lead to a high pass ratio, which is consistent with our findings in the paper. We have revised the manuscript accordingly. Thank you again for the valuable insights.
>
> | Method | Pass Ratio(%) (Qwen2.5) | Pass Ratio(%) (Gemma3) | Ground Ratio(%) (Qwen2.5) | Ground Ratio(%) (Gemma3) |
> | :--- | :---: | :---: | :---: | :---: |
> | in trajectories | 57.50 | 70.00 | 52.49 | 86.69 |
> | in system | 60.00 | **75.00** | 57.41 | 88.10 |
> | in user | **65.00** | **75.00** | 62.78 | **89.79** |
> | in user w/o exp. | 62.50 | 72.50 | **71.24** | 84.79 |
>
>
>
> > Q1. Will more delibrate prompting-based LLM agent baselines perform as well as adding this GNN-based prediction module?
>
> Thanks for the valuable question. In general, we believe it is hard to use delibrate prompt achieve a higher performance than using our method. Because a delibrate prompt usually requires a strong reasoning ability LLM-Agent. It doesn't perform well on a small LLM. To address this concern, we add another experiment on ReAct and Reflexion. As the results show in below, we observe that the ReAct and Reflexion method have a poor performance than ours. Thanks again for the valuable question. We have added the experiments in the Appendix.
>
> | Method | Pass Ratio(%) (Qwen2.5) | Pass Ratio(%) (Gemma3) | Ground Ratio(%) (Qwen2.5) | Ground Ratio(%) (Gemma3) |
> | :--- | :---: | :---: | :---: | :---: |
> | ReAct | 62.50 | 70.00 | 69.86 | 86.60 |
> | Reflextion | 60.00 | 67.50 | **71.18** | 79.60 |
> | Ours | **65.00** | **75.00** | 62.78 | **89.78** |
>
>
> We thank you again for your constructive comments and for your efforts to improve the quality of our paper.

---

### Author Response · Authors · 2025-11-26
**Global Response to Reviewers**

Dear Reviewers,

We thank all reviewers for the constructive and valuable feedback. In response, we have made substantial revisions to improve the clarity, structure, and empirical validation of our work. Below, we summarize the major updates incorporated into the revised manuscript:

- We have added the motivation for the label space in the Preliminaries and provided an example to illustrate how labels are assigned.

- To clarify how the dataset is constructed, we detailed the use of golden-path trajectories in Appendix B.2.

- In Section 3.1 (Graph Construction), we now explain why we adopt an action-centric graph design grounded in the POMDP framework. In Section 3.2, we corrected notation errors and added the appropriate references for the GNN modules.

- In the Related Work section, we included additional graph-related research, even though it lies beyond the primary scope of this paper.

- In Appendix C, we added variance analyses for both the detection performance and the feedback evaluation. To better compare with existing approaches, we also consider comparisons with ReAct and Reflexion.

- For the ablation study, we added experiments on the relationship between detection accuracy and performance improvement, and we conducted additional experiments examining the effects of different feedback-integration mechanisms.

We are grateful for the reviewers’ feedback, which has significantly improved the quality of our work. We hope that the clarifications and additional analyses provided above will assist in your final assessment of our submission.

---

### Author Response · Authors · 2025-12-03
**Summary of Rebuttal and Contributions for Paper**

Dear Area Chair,

We sincerely appreciate your dedication to evaluating our submission and rebuttal. The additional reviewing responsibilities undertaken by Area Chairs are invaluable, and we acknowledge the complexities of ensuring equitable assessments under such demanding conditions. We respectfully submit for your consideration: (1) a synopsis of our contributions, (2) an overview of how our rebuttal addressed reviewers' main concerns.

**Main contribution of this paper:**
- Grounded in the Partially Observable Markov Decision Process (POMDP) formulation, we define an environment-irrelevant action label space for LLM-agent decision-making. Based on this new label space, we introduce GEBUGGER, a probabilistic graph–based method designed to detect step-level action errors in LLM agents.
- We propose TRAJECTORY GRAPH COPILOT, a framework that integrates this error-detection module as an independent component. Acting as a graph-based diagnostic tool, it enhances LLM-agent performance by providing real-time error warnings.
- Experiments across multiple environments and multiple LLM agents demonstrate that GEBUGGER achieves superior error-detection performance and that the overall TRAJECTORY GRAPH COPILOT framework consistently improves agent behavior.


**Address the Reviewers' Main Concerns:**

1. **_Stronger Baselines (Reviewers: ZAio, 6EJw, V4Co)_**

      We have added baselines include two categories: (1) error detection baselines and (2) agent feedback framework baselines.
      - We added stronger backbones, including Qwen2.5-0.5B and TextGCN as a graph-based baseline. Across all settings, our method shows clear improvements.
      - We added comparisons between ours and established agent-improvement frameworks such as ReAct and Reflexion. Results show that ours outperforms both in overall task success.


2. **_Robustness, Variance, and Sensitivity Analysis(Reviewers: ZAio, V4Co)_**

     We have added experiments to provide results with error bars (mean ± std dev) for both error detection and feedback evaluation on TextWorld, showing consistent performance trends. We also conducted an ablation study by injecting random noise into the error detection results. The performance degraded gracefully as noise increased, validating the robustness and importance of the detection module.

3. **_Data Transparency and Feedback Integration_(Reviewers: 6EJw, ZAio)**

      - In this paper, we have collected the golden path from the environment to serve as the reference for action label annotation. The raw data, processed data, and code will be public available, once the paper is accepted.
     - For the feedback integration methods, we have added an ablation study comparing four settings in Appendix, finding that user-prompt integration generally works best.


4. **_Theoretical Assumptions and Generalization(Reviewers: ZAio, 6EJw, B51S, V4Co)_**

      - Our theoretical assumptions are based on the Markov Blanket, which aligns with our POMDP based action error label space. We provide an example in the Appendix illustrating how an action error can be determined by tracing back only a limited number of steps. We also validated Assumption 2 empirically by using a stronger backbone (Qwen2.5-0.5B), showing that graph-based methods continue to outperform sequence-based methods even with improved context modeling.
      - Regarding generalization, our goal is to learn domain knowledge and provide action-level feedback. Our experiments already cover OOD scenarios, including unseen tasks/rooms in ALFWorld and unseen environments in ScienceWorld. Broader cross-environment generalization is beyond the scope of this work and is an interesting direction for future study.




We are grateful for the reviewers’ feedback, which has significantly improved the quality of our work. We are encouraged that Reviewer B51S responded positively to these clarifications and adjusted the average score.


Thank you again for your time and careful consideration of our work. We hope this summary, our revised manuscript, and rebuttal will be helpful for your evaluation.

Best,

The Authors

---

### Meta-Review · Area_Chair_WKL5 · 2026-01-07

**Summary:**

This paper proposes a framework to diagnose possible action errors before executing them with the goal of improving LLM agents' performance on long-horizon tasks. The work models past trajectories as a probabilistic graph and uses a graph neural net to detect actions that can result in errors. These are provided to the agents for self-correction, not requiring fine-tuning and lead to improvements in success rates on multiple agentic benchmarks.

**Reviewer Concerns:**

The work is interesting, however, first the approach requires careful and costly annotations and hence the work may not scale to other frameworks. Authors mention experimentation is also done on previously unseen test sets, but domain transfer is not considered. Second, reviewers ask several clarification questions. These are mainly answered in the rebuttals, but given the amount of such exchanges, it may be best to re-review the paper once the clarifications are integrated.

**Reviewer Scores:**

I would expect that one of the reviewers (B51S) may increase their scores and not others.

---

### Decision · Program_Chairs · 2026-01-26

Reject